



# Severe hail detection with C-band dual-polarisation radars using convolutional neural networks

Vincent Forcadell[1, 4], Clotilde Augros[1], Olivier Caumont[1, 2], Kevin Dedieu[4], Maxandre Ouradou[1], Cloé David[1], Jordi Figueras i Ventura[3], Olivier Laurantin[3], and Hassan Al-Sakka[5]

[1]CNRM, Université de Toulouse, Météo-France, CNRS, Toulouse, France
[2]Météo-France, Direction des opérations pour la prévision, Toulouse, France
[3]Météo-France, Direction des systèmes d'observation, Centre de Météorologie Radar, Toulouse, France
[4]Descartes Underwriting, Paris, France
[5]Leonardo Germany GmbH, Neuss, Germany

**Correspondence:** Vincent Forcadell (vincent.forcadell@gmail.com)

**Abstract.** Radar has consistently proven to be the most reliable source of information for the remote detection of hail within storms in real-time. Currently, existing hail detection techniques have limited ability to clearly distinguish storms that produce severe hail from those that do not. This often results in a prohibitive number of false alarms that hamper real-time decision-making. This study utilises convolutional neural network (CNN) models trained on dual-polarisation radar data to detect severe

hail occurrence on the ground. The morphology of the storms is studied by leveraging the capabilities of a CNN. A database of images of $60\,\text{km} \times 60\,\text{km}$ containing 19 different radar-derived features is built above severe hail reports ($\geq 2\,\text{cm}$) and above rain or small hail reports (rain or hail $< 2\,\text{cm}$) created for the occasion with the help of a cell-identification algorithm. After a tuning phase on the CNN architecture and its input size, the CNN is trained to output one probability of severe hail on the ground per image of $30\,\text{km} \times 30\,\text{km}$. A test set of 1396 images between 2018 and 2023 demonstrates that the CNN

method outperforms state-of-the-art methods according to various metrics. A feature importance study indicates that existing hail proxies as input features are beneficial to the CNN, particularly the maximum estimated size of hail (MESH). The study demonstrates that many of the existing radar hail proxies can be adjusted using a threshold value and a threshold area to achieve similar performance to that of the CNN for severe hail detection. Finally, the output of ten fitted CNN models in inference mode on a hail event is shown.

## 1 Introduction

Hailstorms are severe weather phenomena that pose significant risks to agriculture, infrastructure, and human safety. Accurate detection and monitoring of hail is crucial for issuing timely warnings and minimizing potential damages, as well as assisting damage surveys after an event. Weather surveillance radar systems have proven to be valuable tools for detecting hail (Ryzhkov



and Zrnic, 2019). Dual-polarisation radars use horizontally and vertically polarised electromagnetic waves transmitted to the atmosphere in pulses using a rotating antenna. The echoes returned from targets are analysed to compute various variables within the scanned volume. This data is used to enhance the capabilities of radar systems in detecting and warning about the formation of hail-bearing storms in real-time.

Radar-based hail detection techniques can be divided into two distinct groups. The first group is based on reflectivity at

horizontal polarisation ($Z_H$). Dry hailstones typically exhibit high $Z_H$ values, although they are weaker than those of raindrops of the same size due to a higher dielectric constant for rain (Ryzhkov and Zrnic, 2019). However, due to the fact that for a given content, hail exhibit a particle size distribution that is shifted towards larger diameters in comparison to rain, the reflectivities of dry hail are larger than those of rain for an equivalent content. Melting hail is associated to even larger reflectivities due to an increase of the dielectric constant compared to dry hail, because of the presence of liquid water on its surface (Ryzhkov

et al., 2013b; Ryzhkov and Zrnic, 2019). By analysing $Z_H$ data, either alone or with temperature profiles, meteorologists have attempted to identify the presence of hail and severe hail ($\geq 2\,\mathrm{cm}$). For example, Waldvogel et al. (1979) developed a criterion that combines echo tops (ET), i.e. the maximum height at which the reflectivity reaches a certain value, and the height of the melting layer, to compute a probability of hail (POH). This criterion is still used in several European countries as a proxy for hail occurrence (Delobbe and Holleman, 2006; Foote et al., 2005; Trefalt et al., 2023). In an effort to utilise this vertical

information in storms, studies have sought to produce proxies that integrate reflectivity over the vertical, such as the Vertically Integrated Liquid (VIL, Greene and Clark, 1972; Pilorz et al., 2022) and the VIL density (VILd, Amburn and Wolf, 1997). Since hail mainly forms within storm updrafts and above the melting layer, relationships between vertically integrated $Z_H$ values and temperature profiles have been developed for hail and severe hail detection (Witt et al., 1998; Trefalt et al., 2023; Murillo and Homeyer, 2019). Among these methods, some are based on the severe hail index (SHI) developed by Witt et al.

(1998). The SHI is derived from the weighted integral of reflectivity over the vertical, where values are weighted based on their relative position to the hail growth zone. Several proxies, such as the probability of severe hail (POSH) and the maximum estimated size of hail (MESH) were developed upon it (Witt et al., 1998). These aforementioned methods using $Z_H$ as a main variable are still widely used operationally in weather services, either for real-time applications (Smith et al., 2016) or for the production of hail climatologies (US: Wang et al., 2018, Australia: Soderholm et al., 2017; Brook et al., 2024, Switzerland:

Nisi et al., 2020). While providing a high probability of detection depending on the validation methodology, these techniques are known to suffer from false alarms (Holleman, 2001; Ortega, 2021; Pilorz et al., 2022).

The second group of techniques uses dual-polarisation radar data, also called polarimetric data, which provides valuable information about the shape of targets and the precipitation type (Zrnić et al., 1993; Vivekanandan et al., 1999; Kumjian, 2013a, b; Ryzhkov et al., 2013a; Ryzhkov and Zrnic, 2019). Polarimetric radars allow the computation of new variables: the

differential reflectivity ($Z_{DR}$), the copolar correlation coefficient, also called cross-correlation coefficient ($\rho_{HV}$), and the specific differential phase ($K_{DP}$). As polarimetric variables distributions can overlap significantly among different precipitation types (Kumjian, 2013a), a fuzzy-logic scheme appeared well-suited to answer the problem of classification of radar echoes (Vivekanandan et al., 1999), where hail could be detected as an independent class. A fuzzy-logic algorithm is based on assigning each precipitation type its own range of values for single and dual-polarisation variables. These ranges are determined



through simulations or physical interpretations of the radar variables (Park et al., 2009; Ryzhkov et al., 2013b; Kumjian, 2013a). The grade of membership to a particular type being within the radar gate, given the value of a variable, is computed using a membership function, typically trapezoidal. The aggregation of the membership grades of each precipitation type for each radar variable enables the determination of the most dominant precipitation type within the radar gate (Kumjian, 2013a). Based on this principle, a significant number of fuzzy-logic algorithms using dual-polarisation were developed (Vivekanandan

et al., 1999; Straka et al., 2000; Gourley et al., 2007; Al-Sakka et al., 2013; Ryzhkov et al., 2013b; Ortega et al., 2016; Steinert et al., 2021). For hail, due to the wide distribution of possible axis ratios and hailstone shapes in real conditions (Soderholm and Kumjian, 2023; Giammanco et al., 2017), there is a significant increase in the variability of the scattering properties, particularly at C-band due to resonance scattering at large sizes. This may prevent a good discrimination between hail and other precipitation types using a fuzzy-logic approach based solely on membership hypotheses of polarimetric variables (Jiang

et al., 2019; Shedd et al., 2021). Furthermore, classes of hail within fuzzy-logic algorithms are difficult to validate given the scarcity of hail reports available both on the ground and aloft (Al-Sakka et al., 2013; Ortega et al., 2016). Despite these limitations, radar-based fuzzy-logic classification remains the best method for discriminating hail from other types of precipitation (Kumjian, 2013b; Ortega, 2013).

   The common limitation of the aforementioned single- and dual-polarisation hail detection techniques is the fact that they

are computed on a pixel-by-pixel or column-by-column basis. They can be represented as functions mapped to all radar pixels coming either from the volumetric radar data or deduced from the vertical integration of radar variables. These pixel-based methods do not allow the broader view of the radar variables, their spatial structure and the morphology of the storm to be studied. Additionally, the models are unable to accurately represent potential intricate and non-linear relationships between model variables or radar variables and hail on the ground. To tackle these limitations, techniques capable of 1) harnessing

the morphology of spatially-coherent features within radar images or 2) studying the intricate relationships between radar or environmental variables and ground truth were developed. In recent years, machine learning and deep learning radar hail detection techniques have gained traction. In the work of Wang et al. (2018), they developed a convolutional neural network (CNN, Lecun et al., 1998) applied to three-dimensional reflectivity grids in order to detect hail. Using $70\,\text{km} \times 70\,\text{km}$ reflectivity images at different altitudes centered on the cell cores, they showed better discrimination of hail compared to the POSH

method, particularly reducing the number of false alarms. In the work of Shi et al. (2020), they tracked convective cells and trained a bagging class-weighted support-vector machine (CWSVM) using single-polarisation cell-based features and environmental information from proximity soundings. By comparing with common reflectivity based hail proxies, they showed better performances for their fitted model. Finally, in the work of Ackermann et al. (2024), they trained a neural network using the severe hail index (SHI, Witt et al., 1998) and variables from ERA5 (Hersbach et al., 2020) to estimate the magnitude of

the damage generated by hail on the ground. Using insurance data as ground-truth, they developed a hail damage estimate variable that showed high accuracy on the estimation of damage and its intensity. These prior machine learning and deep learning studies have demonstrated the potential of these techniques to partially address the lack of information on hail growth processes. Consequently, the consideration of hail detection as an image-based problem where the morphology of storms can be taken into account seems a promising approach to enhance the hail detection capabilities of radar networks.





**Table 1.** Example of a super-cycle for the radar of Toulouse. The $90°$ elevation angle is used for $Z_{DR}$ calibration.

| sub-cycle | Elevation angles | | | | | |
|---|---|---|---|---|---|---|
| 0 min | 90° | 8.5° | 5.5° | | | |
| 5 min | 10.5° | 7.5° | 4.5° | 2.5° | 1.5° | 0.8° |
| 10 min | 9.5° | 6.5° | 3.5° | | | |

This study aims to train different CNN models for the detection of severe hail ($\geq 2\,\mathrm{cm}$) on the ground using polarimetric radar data. Although studies have already explored the use of CNNs for hail occurrence detection, to the authors' knowledge, none have attempted to use radar polarimetric variables for severe hail detection with CNNs. The framework developed herein is based on a dataset of severe hail cases ($\geq 2\,\mathrm{cm}$) and negative cases including rain or small hail ($< 2\,\mathrm{cm}$). First, the data gathered for this study are presented in section 2. Then, the methods explaining the features, the tuning phase to choose the

CNN's architecture and its input size, and the metrics are described in section 3. Finally, the results presented in section 4 are divided into four parts: 1) the results of the tuning phase (section 4.1), 2) the feature selection and feature importance studies (section 4.2), and 3) a comparison with state-of-the-art (section 4.3). Finally, the conclusions of this study present a summary of the contributions made to the field of severe hail detection and suggest potential applications for future research.

## 2    Data

### 2.1    Radar

This study uses data from C-band radars within metropolitan France (Fig. 1). The volume coverage pattern (VCP) of each radar consists of super-cycles of $15\,\mathrm{min}$ in which five to seven elevation angles are scanned, depending on the radar (Table 1). Each $15\,\mathrm{min}$ super-cycle contains three $5\,\mathrm{min}$ sub-cycles with the three lowest elevation angles remaining the same and the upper elevation angles changing every $5\,\mathrm{min}$. The raw volumetric radar data, with a range resolution of $240\,\mathrm{m}$ and an azimuthal

sampling of $0.5°$, are processed through a polarimetric processing chain (Figueras i Ventura et al., 2012). Non-meteorological echoes are removed, partial beam blockage is corrected, and $Z_H$ and $Z_{DR}$ are corrected from attenuation (Gourley et al., 2007; Figueras i Ventura et al., 2012; Figureas i Ventura and Tabary, 2013). Volumetric radar data are not corrected for advection between successive elevation angles. The corrected radar data was collected above severe hail reports (see section 2.3) and above rain or small hail reports (see section 2.4) to provide the radar images fed to the deep learning framework. Polarimetric

radar variables considered in this study are $Z_H$, $Z_{DR}$, $K_{DP}$ and $\rho_{HV}$.

### 2.2    Storm-cell identification

Two independent storm-cell identification algorithms are used in this study. The first cell-identification algorithm is used to assist in the production of rain or small hail reports ($< 2\,\mathrm{cm}$). The algorithm is adapted from Morel and Sénési (2002), and





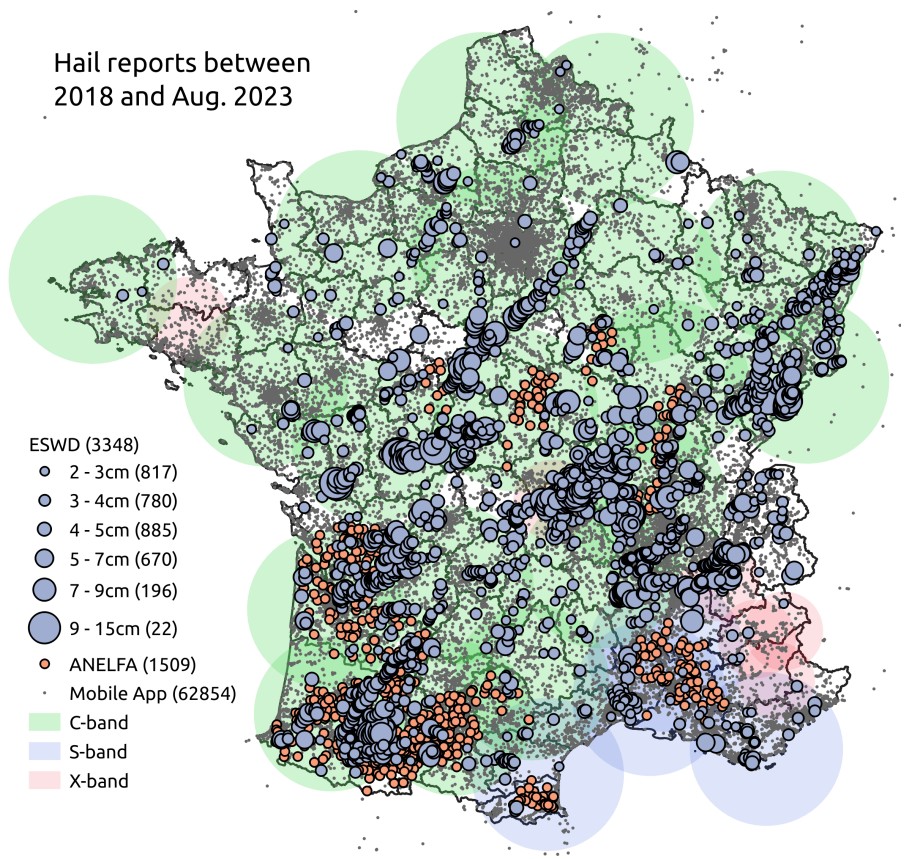

**Figure 1.** Hail reports between 2018 and August 2023 from the ESWD (grey), the hail pad network of the ANELFA (orange) and the mobile application of Météo-France (small black dots).

applied to the national reflectivity composite product available every 5 min at a 1 km horizontal resolution (Caumont et al.,
2021). The algorithm defines cells as a contiguous set of pixels above a certain reflectivity threshold. Cell objects with four different thresholds are defined: 36 dBZ, 42 dBZ, 48 dBZ and 56 dBZ. Cell splits and merges are managed by comparing cell overlaps between consecutive images, taking into account cell motion (Morel and Sénési, 2002). This basic thresholding scheme allows a fast computation of cell cores with different degrees of severity, but can suffer from discontinuity in time compared to more sophisticated algorithms (Lakshmanan et al., 2009).

A more advanced cell-tracking algorithm was employed on a single event to illustrate the inference process for the methods developed herein. The use of a different cell tracking algorithm for inference is necessary because the former algorithm is not always able to accurately locate cell centroids. In the first cell tracking algorithm, centroids are defined as the geometric mean within the contours and are not weighted by the reflectivity values within the cell. As a result, centroids may not be within the cell core, but far away from it, preventing continuous tracking of cells every 5 min. The more sophisticated cell-
tracking algorithm is based on the open-source Python package *tobac* (Heikenfeld et al., 2019). It comprises a toolbox where





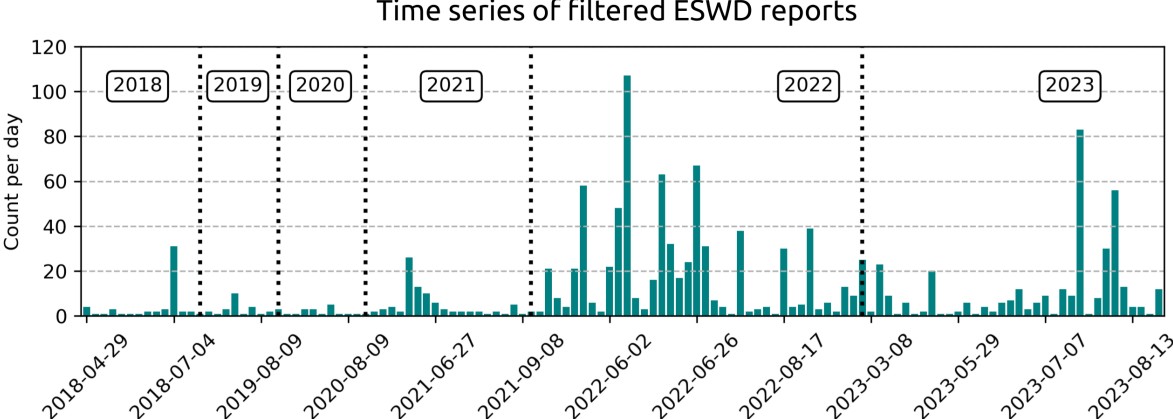

**Figure 2.** Time series of the 1169 filtered ESWD severe hail reports ($\geq 2\,\mathrm{cm}$) used in this study.

cell tracking and segmentation algorithms can be applied using different parameters. In this study, the cell tracking feature is employed exclusively. Cells are identified within the national composite reflectivity as one or more contiguous regions of reflectivity values that meet or exceed a threshold. The thresholds used in this study are $36\,\mathrm{dBZ}$, $42\,\mathrm{dBZ}$ and $48\,\mathrm{dBZ}$. Additional parameters are used to set a minimum cell size per threshold: $30\,\mathrm{km}^2$, $10\,\mathrm{km}^2$ and $2\,\mathrm{km}^2$ respectively. As multiple

reflectivity thresholds are specified, the centroid of each $42\,\mathrm{dBZ}$ cell that exist within a $36\,\mathrm{dBZ}$ region supersede and replace the centroid detected for the encompassing $36\,\mathrm{dBZ}$ cell, as explained in Heikenfeld et al. (2019). The combination of different thresholds allows for the detection of cell centroids for cells at their initial or decay stage, as well as the identification of cell cores during the mature stage.

## 2.3   Severe-hail reports

This study utilises various sources of hail reports, either as ground truth for severe hail or to assist in constructing the rain or small hail database. The European Severe Weather Database (ESWD, Dotzek et al., 2009), an initiative of the European Severe Storm Laboratory (ESSL), is the primary source of severe hail reports used in this study. Severe weather phenomena are reported by volunteer observers, weather services, or individuals and are quality controlled by the ESSL into four levels of quality, ranging from QC0 to QC4 (Groenemeijer and Kühne, 2014). To localise and estimate the maximum hail size, images

from social media or local newspapers are frequently used. From January 2018 to August 2023, the ESWD collected 3348 reports in France with a maximum hail size information above $2\,\mathrm{cm}$ (Fig. 1). Although the ESWD management team applies quality checks to the reports, errors in the hailfall time or report localisation may still occur. To reduce their impact, the hailfall time was adjusted by examining the reflectivities from the nearest radar within a time range of $\pm30\,\mathrm{min}$. If needed, the report time was shifted to the time when a storm cell passed over the report. If multiple cells were observed over the report within

the time range, the time of the closest cell to the reported time was retained. If no cell was detected within that time frame,



the report was discarded. A significant proportion of reports produced by the same storm at the same time remains in the database. It artificially increases the number of independent storm cells that produced severe hail. To avoid duplicating radar images centered on reports that are really close to each other, a density-based clustering algorithm (DBSCAN, Ester et al., 1996) is applied to find reports within $10\,\mathrm{km}$ to each other every $5\,\mathrm{min}$. The report that is the closest to the barycenter of

collected reports is kept. The total number of severe hail reports used for training decreased from 3348 to 1169. Fig. 2 shows their distribution over time. The filtered ESWD reports are considered the only trustable source of severe hail reports for the remainder of the study.

The study also collected 1509 hail pad reports between 2018 and 2022, purchased from the Association Nationale d'Étude et de Lutte contre les Fléaux Atmosphériques (ANELFA, Dessens et al., 2007). Its network of hail pads covers most of the

south-west of France (Fig. 1). A hail pad consist of a $30\,\mathrm{cm} \times 50\,\mathrm{cm} \times 7\,\mathrm{cm}$ layer of polyester placed on the ground or mounted on a pole. Hail reports are generated from photographs of hail pads after hailstorms and are processed by the ANELFA using computer vision techniques to infer hail characteristics. There is only one report per day per hail pad, and each report is accompanied by an estimated time of hail fall by the observer. Numerous quantities are available in the reports, such as maximum diameter or hail size distribution. The main challenge with hail pad data is the small sampling area of the pad, which

prevents accurate measurement of maximum hailstone size, as the largest hailstone can easily be missed (Smith and Waldvogel, 1989). The possible systematic underestimation of the maximum diameter due to this sampling error makes it impossible to consider such data as severe hail reports above $2\,\mathrm{cm}$. However, they remain important for the construction of the database of rain or small hail reports.

Hail reports were also collected through the crowdsourcing feature of Météo-France's mobile application between 2018 and

August 2023. The application allows users to report weather events such as snow, strong winds and hail, which are then located using GPS technology embedded in mobile phones. Since 2014, users can add information about the size of the hailstones and include a picture. The hail size categories available are a) lower than $0.5\,\mathrm{cm}$, b) $0.5\,\mathrm{cm}$ to $1.0\,\mathrm{cm}$, c) $1.0\,\mathrm{cm}$ to $2.0\,\mathrm{cm}$, d) greater than $2.0\,\mathrm{cm}$. A large quantity of hail is reported between 2018 and August 2023 (137,108 reports). However, the database may contain a significant misrepresentation of hail occurrence due to the lack of systematic quality controls. Observers may report

hail despite the absence of reflectivity data indicating precipitation, or there may be potential errors in space and time caused by people reporting hail after it has fallen. To correct for possible biases, a consistency check was carried out. Cell-objects of $42\,\mathrm{dBZ}$ from the first cell identification algorithm (section 2.2) were collected within a time period of $-120\,\mathrm{min}$ to $+30\,\mathrm{min}$ around each report. If the distance between the report and the nearest $42\,\mathrm{dBZ}$ cell within that period was more than $15\,\mathrm{km}$, the report was discarded. The $42\,\mathrm{dBZ}$ reflectivity threshold was chosen because small and melting hail above $5\,\mathrm{mm}$ is hardly

reported at reflectivity values lower than $45\,\mathrm{dBZ}$ (Ryzhkov and Zrnic, 2019). The selected time interval is needed to consider potential delays between the reported time and the actual hailfall time. A delay of two hours prior to the reported time was deemed adequate to account for this. Finally, a distance of $15\,\mathrm{km}$ between a report and the nearest $42\,\mathrm{dBZ}$ contour was chosen to represent the median commuting distance travelled by the rural French population each day (INSEE, 2023). Using that consistency check, the quantity of reports decreased from $137\,108$ to $64\,051$, still covering $45\,\%$ of the days within the study.

In certain highly populated areas, the frequency of hail occurrence reported by the application remains significantly higher





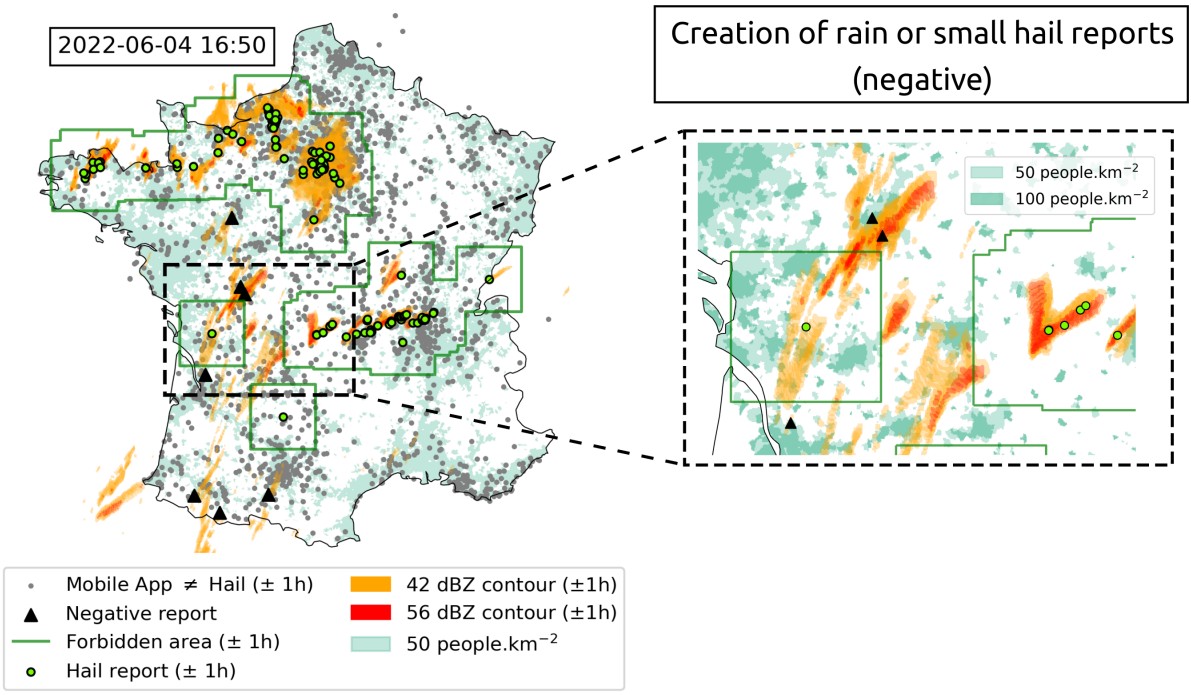

**Figure 3.** Construction of the rain or small-hail reports on the 4th June 2022 at 16:50 (UTC) during a convective outbreak where hail was reported. Green dots represent hail reports (ESWD + Mobile application + ANELFA) within a time interval of ±1 h. Green squares are 'forbidden' areas around hail reports (120 km × 120 km) where a rain or small hail report cannot be created at 16:50 (UTC). The orange and red colours represent the 42 dBZ and 56 dBZ cells cumulative contours within a time interval of ±1 h. Grey dots represent the reports from the application that are not hail reports within a time interval of ±1 h. Light and dark turquoise show populated areas with more than 50 people.km$^{-2}$ and 100 people.km$^{-2}$, respectively. Black triangles represent negative (rain or small hail) reports created at the mentioned timestamp. They represent the intersection of reflectivity contours and areas of more than 100 people.km$^{-2}$ outside forbidden areas. Some of them are discarded based on further filtering explained in section 2.4.

than what known climatologies suggest (Dessens, 1986; Punge et al., 2017). Furthermore, only 28 % of the remaining 64 051 reports contain hail size information, and about 1.1 % is severe hail ($\geq$ 2 cm). Because of the database's size, manual filtering was not possible within the scope of this work. Therefore, the final quality of the collaborative reports remains uncertain. As a result, it was not used as ground truth for severe hail but it assisted the construction of the rain or small hail database.

## 2.4 Rain or small-hail reports

Rain or small hail reports are created as reports produced by either rain or small hail below 2 cm. In order for the CNN to accurately distinguish between radar images that result in severe hail and those that do not, it is crucial that the training set includes instances where severe hail did not occur on the ground. Rain or small hail reports are built to include storms that may



be conducive to hail formation but did not produce severe hail at the ground. The identification of such storms is necessary for
the validation of severe hail detection algorithms. They are considered edge cases and often produce many false alarms with
current hail detection methods, making it difficult for forecasters to distinguish between severe and non-severe hail storms.

Rain and small hail reports were searched every $20\,\mathrm{min}$ during hail seasons (March-September) between 2018 and August
2023. Several precautions were taken to build this database. First, times and locations with no potential for hail formation were
excluded based on a minimum reflectivity threshold. Thus, a disproportionate number of useless cases to train the CNN were
discarded. Second, locations in sparsely populated areas and times of day when hail cannot be reported were excluded, as was
done in the study by Kopp et al. (2024). Finally, entire areas during time intervals around hail reports were forbidden to avoid
domains where severe hail was highly probable. As a result, an initial filtering was applied every $20\,\mathrm{min}$ using cell objects,
where the following locations were kept:

- locations below cell objects that had a maximum $Z_H$ above $45\,\mathrm{dBZ}$.

- locations at the intersection between cell objects and a highly populated area of at least $100\,\mathrm{people\,km^{-2}}$.

- locations within working hours (7:00am-10:00pm).

- locations outside 'forbidden' areas, defined as squares of $120\,\mathrm{km}\times 120\,\mathrm{km}$ around all available hail reports within a time
  interval of $\pm 1\,\mathrm{h}$. The hail reports considered here are a combination of raw severe hail reports from the ESWD (3348),
  hail pad measurements from the ANELFA (1509) and filtered collaborative reports from the Météo-France mobile ap-
  plication (62 854).

An example of the rain or small hail reports produced by such filters applied to a convective outbreak on the 4[th] June 2022 at
16:50 (UTC) is shown in Fig. 3. Using a filter that combines all available hail reports to exclude 'forbidden' areas where rain or
small hail reports cannot be created was considered the best option, given the significant uncertainty in the quality of hail pad
measurements and collaborative reports. However, a risk remains that avoiding such forbidden areas around hail reports may
result in the withdrawal of several small hail cases ($< 2\,\mathrm{cm}$). The filtering assumed that all missed severe hail by the ESWD
database was correctly observed in highly populated areas within working hours by other databases, even with a wrongly
observed hail size, as it attracts more attention from both the media and the public (Punge and Kunz, 2016). This hypothesis
is contingent upon the presence of a sufficient number of individuals capable of recording hail. It can be demonstrated that
a non-negligible number of non-hail observations are produced by the mobile application within the French territory every
two hours (Fig. 3), reducing the risk of missing severe hail. These steps serve to ensure that rain or small hail reports are not
contaminated by severe hail, which is of the utmost importance for the relevance of the method and the interpretation of its
results.

In order to reduce the number of cases that produced moderate $Z_H$ values, an additional filter was applied. Since mild
precipitation events are climatologically predominant compared to severe and extreme precipitation events, they can populate
most of the rain or small hail cases, even if a minimum threshold of $45\,\mathrm{dBZ}$ was set. In order to prevent the CNN from learning





with a disproportionate number of mild cases, a second filter was applied to cases that had cell-objects with a maximum $Z_H$ below $56\,\mathrm{dBZ}$. These cases were divided into two categories: those produced by cells with a maximum $Z_H$ 1) beetween $45\,\mathrm{dBZ}$ and $48\,\mathrm{dBZ}$, and 2) between $48\,\mathrm{dBZ}$ and $56\,\mathrm{dBZ}$. The reports with the largest cell area per bin of $0.2\,\mathrm{dBZ}$ for each category

were then retained. This was done to ensure that rain or small hail reports were produced by large enough storms where hail is plausible, as severe hail is mainly produced in supercell and multicell convective systems (see section 2.5). In the event that reports were situated at a distance of less than $15\,\mathrm{km}$ from one another, only the report produced by the cell exhibiting the highest reflectivity was included. In the event that they originated from the same cell, one was selected at random. This methodology ensured that rain or small hail reports were extracted from independent stages of a storm's life cycle.

After these different steps of filtering, the rain or small hail database contained 2605 reports during hail seasons between 2018 and August 2023. Cell objects formed by the cell identification algorithm were also gathered above the filtered severe hail reports (section 2.1). The fitted probability density functions (PDF) of $\mathrm{max}(Z_H)$ within the cell and the cell area above $56\,\mathrm{dBZ}$ are compared in Fig. 4. Despite the efforts to gather intense storms in the rain or small hail dataset, Fig. 4 shows only a partial overlap between the distributions on both datasets, indicating that the the biggest cases in terms of maximum reflectivity

and cell area were mostly produced by severe hail storms. This behaviour may be a consequence of the storm modes embedded in each dataset, where severe hail is nearly systematically produced by large, intense and highly organised systems such as supercells. A storm mode assessment is performed in section 2.5.

It is crucial to acknowledge that it was not feasible to make sure that small hail was included in the rain or small hail dataset. Indeed, small hail is less likely to be reported by observers, and a significant degree of uncertainty contaminates the existing

databases that have the capacity to report it (Météo-France crowd-sourcing application, ANELFA hail pads). Consequently, it is assumed that by selecting the strongest storm cases outside areas where hail was reported using the aforementioned filter, it was possible to include potential instances of small hail. In the most unfavourable scenario, the rain or small hail database is populated with instances of rain or heavy rain only, which still contributes to the generation of false alarms in existing severe hail detection algorithms.

**2.5 Storm mode**

In order to gain further insight into the database, a storm mode assessment was conducted. The storms responsible for the production of severe hail reports and rain or small hail reports were categorised into four distinct modes: supercell, multicell, isolated cell and unknown. However, it was deemed impractical to label the storms that produced all the reports presented above. Indeed, a certain proportion of the reports were isolated, and manually labelling them would have required too much

time. As a result, only the clusters of reports comprising at least two reports were labelled. For the severe hail reports, all were kept. For the rain or small hail reports, only the most severe with a cell producing a $\mathrm{max}\,Z_H \geq 56\,\mathrm{dBZ}$ were kept. This likely introduces a bias towards more severe storm modes and provides an inaccurate representation of the occurrence of certain storm modes, particularly isolated cells. Nevertheless, it was deemed necessary to examine the data, despite the potential for inaccuracy, in order to ascertain whether a discernible signal existed with regard to specific storm modes in relation to storms

accompanied by severe hail.





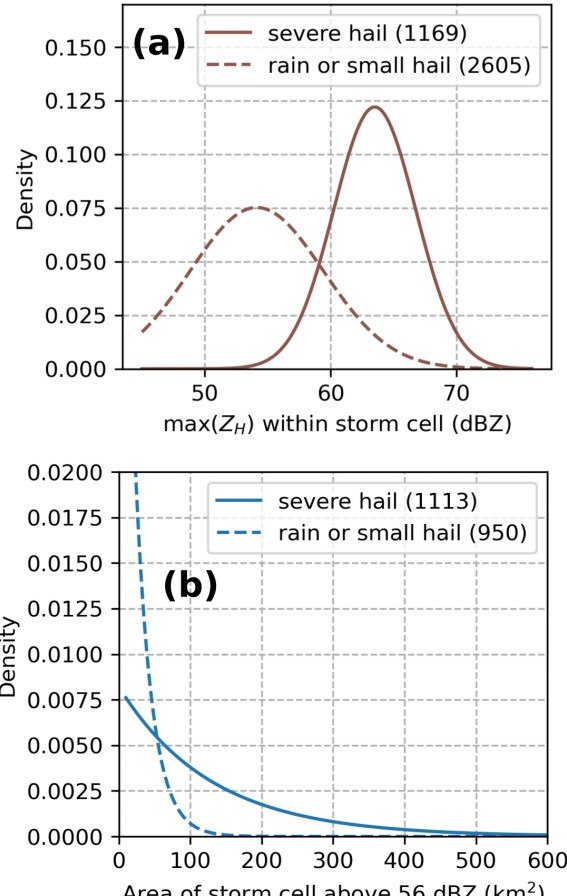

**Figure 4.** Fitted probability density functions (PDF) for storm cell objects identified above severe hail and rain or small hail reports. **(a)** PDF of the maximum reflectivity ($\max(Z_H)$) within storm cells. **(b)** PDF of the area for storm cells with the $56\,\text{dBZ}$ threshold.

The clusters of reports were created using a spatio-temporal DBSCAN algorithm (ST-DBSCAN, Birant and Kut, 2007). The severe hail reports are clustered with $\delta x = 15\,\text{km}$ and $\delta t = 10\,\text{min}$. The rain or small hail reports are clustered with $\delta x = 30\,\text{km}$ and $\delta t = 60\,\text{min}$. A higher spatio-temporal tolerance was selected for the rain or small hail reports, as they are geographically scarcer than the severe hail reports. The national composite reflectivity product (Caumont et al., 2021) and the cells detected by the first cell identification algorithm (Morel and Sénési, 2002) are gathered around $\pm 90\,\text{min}$ before and after the first and the last report of the cluster, respectively. All the data is superimposed in a visualisation tool that enables navigation through time during the life cycle of the storm, facilitating the identification of relevant signatures for labelling. The labelling was performed independently by two meteorologists, and the results were cross validated.

For supercells, typical signatures in the reflectivity composite were searched: a hook echo, a cell splitting, and/or a deviation of the cells to the right (or to the left) of the main flux (Markowski and Richardson, 2011; Houze, 2014). In the event that a




**Table 2.** Storm mode on 224 severe hail storms ($\geq 2\,\mathrm{cm}$) and 113 rain or small hail storms ($< 2\,\mathrm{cm}$).

|  | Severe hail ($\geq 2\,\mathrm{cm}$) | Rain or small hail ($< 2\,\mathrm{cm}$) |
|---|---|---|
| Supercell | 69.9 % | 3.4 % |
| Multicell | 19.3 % | 86.6 % |
| Isolated cell | 4.4 % | 4.3 % |
| Unknown | 6.4 % | 5.7 % |
| Total | **224** | **113** |

clear line of cells was discernible, the cluster was designated as being part of a multicell system. Conversely, if a cell exhibited a brief lifespan and was isolated from any broader convective system, it was classified as an isolated cell. In the absence of any of the aforementioned criteria or in the event that a determination was precluded due to the passage of multiple cells above the cluster in a brief period of time, the cluster was designated as unknown.

A total of 224 severe hail clusters and 113 rain or small hail clusters were labelled. The results are presented in Table 2. Supercells produce 69.9 % of the severe hail on the ground within this study. This shows the predominance of supercells in the production of severe weather compared to other storm modes, which is in accordance with previous studies (Markowski and Richardson, 2011). The rain or small hail dataset is mainly populated by multicell convective systems (86 %) while only 3.4 % were produced by supercells.

The conclusions in this paragraph remain highly entitled to the data used and the portion of cases selected to perform the storm mode assessment.

### 2.6   Cartesian 3D polarimetric grid

The interpolation algorithm implemented within the Python ARM Radar Toolkit (Helmus and Collis, 2016) is used to generate three-dimensional Cartesian grids above each report. Derived two-dimensional fields from the three-dimensional grids are then
used as input features to the CNN. The algorithm produces the grids with a specified resolution of $250\,\mathrm{m} \times 250\,\mathrm{m} \times 500\,\mathrm{m}$ on $60\,\mathrm{km} \times 60\,\mathrm{km} \times 15\,\mathrm{km}$ by interpolating values from the two nearest radars from each report. The value of each grid point is determined by interpolating from the collected radar points within a given radius of influence (ROI). The ROI increases with distance to the radar, and the ROI value for each grid point in the target cartesian grid is determined by the nearest radar. In order to identify the nearest radar points within the specified ROI of a given grid point, a KD-tree algorithm is employed.
The value of the grid point is calculated by summing the collected values, with each value weighted by an inverse distance weighting function defined by Barnes (1964). The three-dimensional grid is generated for $Z_H$, $Z_{DR}$, $K_{DP}$, and $\rho_{HV}$.

To account for the low vertical sampling resolution of the French radars and to avoid discontinuities in the resulting 3D fields, both above the radar and at long range, a minimum radius of influence of $\mathrm{ROI_{min}} = 2000\,\mathrm{m}$ was defined above each radar. This minimum ROI resulted in a smoothing of the fields. A nearest-neighbour interpolation scheme was also tested (not



shown), but produced strong artefacts within the 3D fields such as holes and stripes, preventing its use. As a result, the Barnes interpolation with a minimum ROI of $2000\,\text{m}$ was kept.

## 2.7  Reference algorithms

This section presents the existing radar-based algorithms that are compared with the CNN approach. They are separated in three different kinds. The first algorithm being compared is an updated version of the fuzzy-logic hydrometeor classification

algorithm from Al-Sakka et al. (2013), which is available at S, C, and X bands. The original version of the algorithm discriminates between six different hydrometeor classes using dual-polarisation radar variables and temperature: biological scatters or ground clutter (BS-GC), rain (RA), wet snow (WS), dry snow (DS), icy particles (IC) and hail (HA). A revised version enables the classification of hail into three distinct categories: small hail (SH; $< 0.5\,\text{cm}$), medium hail (MH; $0.5\,\text{cm}$ to $2\,\text{cm}$), and large hail (LH; $> 2\,\text{cm}$). Details on the updated version can be found in appendix A. It is called A13 thereafter.

The second family of algorithms uses the severe hail index (SHI) developed by Witt et al. (1998) to produce two proxies capable of detecting hail: the probability of severe hail (POSH, Witt et al., 1998) and the maximum estimated size of hail (MESH, Witt et al., 1998; Murillo and Homeyer, 2019). The SHI is calculated by the weighted sum of 3D reflectivities over the vertical, based on the position of radar gates to the hail growth zone ($0\,^\circ\text{C}$ and $-20\,^\circ\text{C}$, Witt et al., 1998). The POSH and MESH relationships are defined as follows:

$$\text{POSH} \quad = \quad 29\ln\frac{\text{SHI}}{\text{WT}} + 50, \text{ with WT} = 57.5H_0 - 121 \tag{1}$$

$$\text{MESH} \quad = \quad 2.54 \times \sqrt{\text{SHI}} \tag{2}$$

$$\text{MESH}_{75} \quad = \quad 15.096 \times \text{SHI}^{0.206} \tag{3}$$

$$\text{MESH}_{95} \quad = \quad 22.157 \times \text{SHI}^{0.212} \tag{4}$$

with WT being a warning threshold calibrated for the POSH to produce the best critical success index (CSI) for the U.S. S-band

radars (Witt et al., 1998), $H_0$ being the altitude of freezing in km, MESH coming from Witt et al. (1998) and $\text{MESH}_{75}$ and $\text{MESH}_{95}$ coming from Murillo and Homeyer (2019). The variables are calculated based on the three-dimensional reflectivity grid and the $0\,^\circ\text{C}$ and $-20\,^\circ\text{C}$ altitudes are extracted from the nearest forecast hour within the AROME model (Brousseau et al., 2016). The AROME model provides hourly forecasts with a horizontal resolution of $0.01^\circ$. The isotherms are regridded to the $250\,\text{m} \times 250\,\text{m}$ horizontal resolution of the three-dimensional grid and interpolated in time to the time of the report.

The third family of algorithms is hail detection algorithms based on echo tops, i.e. the maximum altitude at which a reflectivity threshold is reached. The probability of hail (POH) from Delobbe and Holleman (2006) and Foote et al. (2005) are compared in this study and are constructed as follows:

$$\text{POH}_{\text{Delobbe}} \quad = \quad 0.319 + 0.133\Delta H, \tag{5}$$

$$\text{POH}_{\text{Foote}} \quad = \quad -1.20231 + 1.00184\Delta H - 0.17018\Delta H^2 + 0.01086\Delta H^3, \tag{6}$$

where $\Delta H$ is the difference between the echo top at $45\,\text{dBZ}$ (ET45) and $H_0$. Echo tops are computed using the three-dimensional reflectivity grid.



**Table 3.** Input features to the CNN divided in three categories: polarimetry, storm proxy and hail proxy.

| Group | Acronym | Unit | Description |
|---|---|---|---|
| | $Z_H^{\mathrm{max}}$ | dBZ | maximum $Z_H$ over elevations |
| | $Z_{DR}^*$ | dB | collocated $Z_{DR}$ with $Z_H^{\mathrm{max}}$ |
| | $K_{DP}^*$ | $^\circ\,\mathrm{km}^{-1}$ | collocated $K_{DP}$ with $Z_H^{\mathrm{max}}$ |
| | $\rho_{HV}^*$ | | collocated $\rho_{HV}$ with $Z_H^{\mathrm{max}}$ |
| Polarimetry | $Z_H^{2000}$ | dBZ | $Z_H$ at 2000 m |
| | $Z_{DR}^{2000}$ | dB | $Z_{DR}$ at 2000 m |
| | $K_{DP}^{2000}$ | $^\circ\,\mathrm{km}^{-1}$ | $K_{DP}$ at 2000 m |
| | $\rho_{HV}^{2000}$ | | $\rho_{HV}$ at 2000 m |
| | $Z_{DR}$ column | km | $Z_{DR}$ column height |
| Storm proxy | VIL | $\mathrm{kg\,km}^{-2}$ | vertically integrated liquid |
| | ET45 | m | echo-top at $45\,\mathrm{dBZ}$ |
| Environment | $H_0$ | m | altitude of freezing |
| | $\mathrm{POH_{Delobbe}}$ | % | probability of hail from Delobbe and Holleman (2006) |
| | $\mathrm{POH_{Foote}}$ | % | probability of hail from Foote et al. (2005) |
| | POSH | % | probability of severe hail from Witt et al. (1998) |
| | MESH | mm | maximum estimated size of hail from Witt et al. (1998) |
| Hail proxy | $\mathrm{MESH_{75}}$ | mm | 75[th] percentile maximum estimated size of hail from Murillo and Homeyer (2019) |
| | $\mathrm{MESH_{95}}$ | mm | 95[th] percentile maximum estimated size of hail from Murillo and Homeyer (2019) |
| | A13 | | updated hydrometeor classification from Al-Sakka et al. (2013) |

Finally, the maximum reflectivity over the vertical $Z_H^{\mathrm{max}}$ (see section 3.1) is added as a comparison baseline to all the methods compared in this study.

## 3 Methods

This section outlines the experimental design used to evaluate the performance of the CNNs. To align with machine-learning terminology, the term 'radar variable' has been replaced with 'feature'. A feature represents a 2D radar-derived variable that is fed to the CNN.



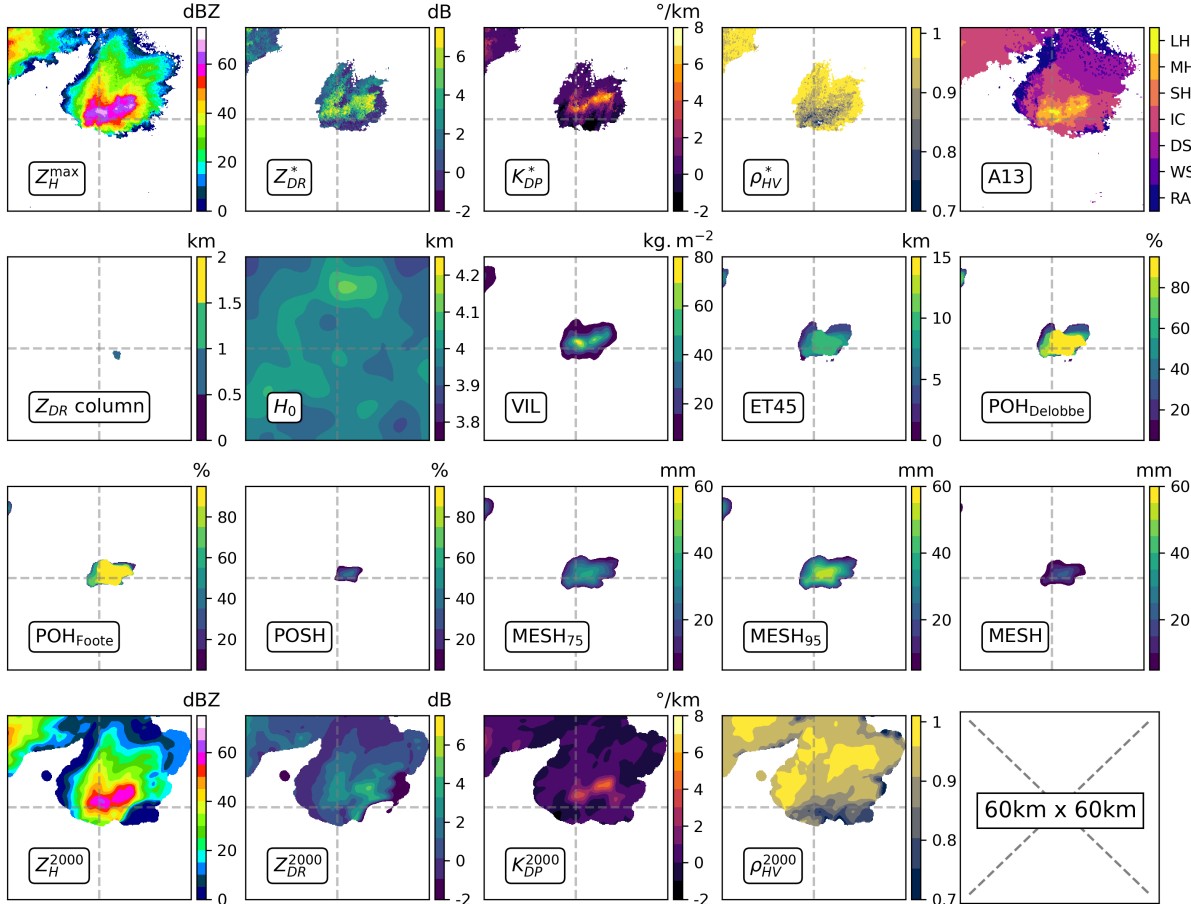

**Figure 5.** Input features defined in Table 3 for a case producing severe hail on the ground. Image size is $60\,\mathrm{km} \times 60\,\mathrm{km}$ and the severe hail report is located at the center of the image.

## 3.1 Input features

For each severe hail report and rain or small hail report, two different sets of inputs are generated: 1) 2D features obtained from the 3D grid, and 2) 2D features extracted directly from the volumetric radar data. Both groups are fed into the CNN. The input features are summarised in Table 3.

The 3D grids are used to generate a number of storm and hail proxies, which are known for their ability to help in the detection of hail. First, the $Z_{DR}$ column is calculated from the 3D grid to account for potential hail formation processes above the freezing level, as it indicates regions with high concentrations of supercooled water and graupel, which are essential for hail growth (Kumjian, 2013b; Kuster et al., 2019). The $Z_{DR}$ column height was calculated using the 3D polarimetric grid, with candidate pixels that met the following criteria: $Z_H \geq 25\,\mathrm{dBZ}$ and $Z_{DR} \geq 2\,\mathrm{dB}$. The height of a column of adjacent candidate pixels is computed as the $Z_{DR}$ column height. A criterion was applied to ensure the continuity of the column above and





below $H_0$ in the event that $500\,\mathrm{m}$ portions of the column were missing in the middle of two candidate pixels over the vertical.
Other 2D input features derived from 3D grids include vertically integrated liquid (VIL, Greene and Clark, 1972), ET45, and
$H_0$. Furthermore, polarimetric features at an altitude of $2\,\mathrm{km}$ are incorporated to account for hail-related signatures at low
altitudes below the altitude of freezing. The $2\,\mathrm{km}$ height was selected as a compromise to achieve optimal 3D radar coverage
while remaining below the freezing level in the majority of cases. It is notable that low $Z_{DR}$ values may be indicative of dry
spherical hail. High $Z_{DR}$ and $K_{DP}$ may suggest the presence of either rain or a mixture of rain and melting hail (Ryzhkov
and Zrnic, 2019). The features at $2\,\mathrm{km}$ include $Z_H^{2000}$, $Z_{DR}^{2000}$, $K_{DP}^{2000}$ an $\rho_{HV}^{2000}$. Finally, a series of hail proxies were subjected
to testing as input features, with the objective of determining the extent to which they might provide additional information
within the framework of a CNN: MESH, MESH$_{75}$, MESH$_{95}$, POSH, POH$_{\mathrm{Foote}}$ and POH$_{\mathrm{Delobbe}}$. A sample of all input features
for a case that resulted in severe hail on the ground is shown in Fig. 5.

The utilisation of 3D interpolation may result in the loss of information present in these features, as it reduces the texture of
the fields (Fig. 5). In order to more accurately represent the native resolution of volumetric radar data, 2D features derived from
volumetric radar data are incorporated in addition to those derived from the 3D grid. Nearest-neighbor interpolation is employed
on the volumetric data at every elevation angle in order to match the horizontal resolution of the 3D grid ($250\,\mathrm{m} \times 250\,\mathrm{m}$). The
nearest-neighbour interpolation is performed separately for each report and for the two nearest radars. In order to account for
the low vertical sampling of French radars and the frequent partial beam blockage at low elevations, 2D features are created
from the interpolated elevations. The initial feature to be considered is the maximum $Z_H$ value over the vertical ($Z_H^{\max}$). The
other ones are called 'collocated' polarimetric features, named respectively $Z_{DR}^*$, $K_{DP}^*$ and $\rho_{HV}^*$. They are selected where
$Z_H^{\max}$ is reached over the elevations. As hail is always detected in areas of high $Z_H$ (Kumjian, 2013a; Ryzhkov and Zrnic,
2019), it appears appropriate to examine the polarimetric signatures where reflectivity is the highest. One disadvantage of this
approach is that the resulting collocated features (2D images) may contain pixels located at different altitudes, which makes
it challenging to interpret their values. To eliminate collocated polarimetric features produced at very high altitudes and low
$Z_H^{\max}$ values, only collocated values where $Z_H^{\max}$ was above $30\,\mathrm{dBZ}$ were retained.

For each report, either severe hail reports or rain or small hail reports, two samples were created, each containing 2D features.
One sample was created for the nearest radar, and the other was created for the second-nearest radar. Both samples share 2D
features that originate from the 3D grid. However, they differ in their $Z_H^{\max}$ and collocated features, as they were produced
independently for each radar. This process helped to augment the dataset, which is considered crucial, particularly given the
scarcity of severe hail reports.

A total of 7523 radar samples were produced from the 1169 severe hail reports and the 2605 rain or small hail reports,
comprising 2335 severe hail and 5188 rain or small hail cases. Fig. 6 illustrates the distributions of maximum values within
samples for a selection of features. It should be noted that the distribution of the maximum reflectivity values within the
images may differ from the distributions obtained with the cell identification algorithm (Fig. 4), as the reflectivity values do not
originate from the same methodology. In the context of this study, distributions of the maximum of input features, including
VIL, ET45, MESH proxies and POSH, exhibit a certain separation between cases of severe hail and those of rain or small hail
(Fig. 6). This may provide insight into the discriminative power of these features for severe hail detection.



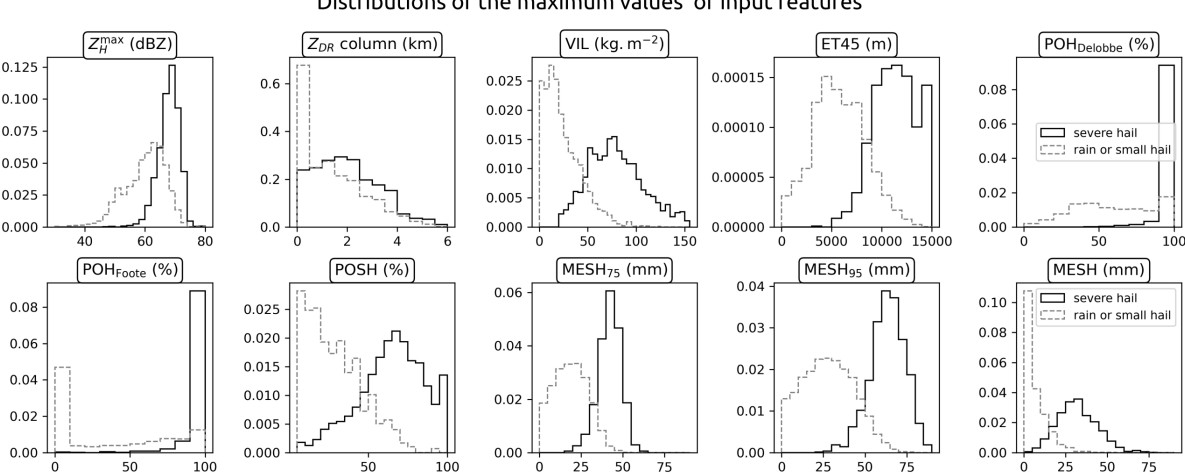

**Figure 6.** Distributions of the maximum value over $60\,\mathrm{km} \times 60\,\mathrm{km}$ images for most of the input features in the severe hail dataset and the rain or small hail dataset.

To analyse the polarimetric variables, the bivariate distributions of $Z_H^{\max}$ and $Z_{DR}^*$ are presented in Fig. 7. The distribution of severe hail exhibits a high density of values with $Z_H^{\max}$ above $50\,\mathrm{dBZ}$ and $Z_{DR}^* \approx 0\,\mathrm{dBZ}$, in accordance with the expected

behaviour of spherical hailstones (Kumjian, 2013a). For rain and small hail cases, $Z_{DR}^*$ increases with $Z_H^{\max}$, as the database may be populated by storms producing either rain or small melting hail that have higher $Z_{DR}$ values compared to larger hail due to a higher dielectric constant for water (Kumjian, 2013a; Ryzhkov and Zrnic, 2019).

### 3.2 Tuning architecture and input size

Two distinct types of CNN architectures are evaluated to identify the optimal architecture and input size. The first type of

architecture is a feed-forward CNN, which draws inspiration from the AlexNet architecture (Krizhevsky et al., 2017). Two models were created from it: the SmallConvNet and the ConvNet. The former comprises only one convolutional layer, while the latter is a deeper architecture with three convolutional layers (Fig. 8). The second kind of architectures tested in this study is a residual network architecture (ResNet, He et al., 2015). The 18-layer variant of the ResNet is used and includes 18 layers of convolutions with skipped connections that increase the accuracy of the network (He et al., 2015). Four input sizes are tested

with the different models: $5\,\mathrm{km} \times 5\,\mathrm{km}$, $15\,\mathrm{km} \times 15\,\mathrm{km}$, $30\,\mathrm{km} \times 30\,\mathrm{km}$ and $50\,\mathrm{km} \times 50\,\mathrm{km}$. Every combination of model and input size is trained, and the combination that yields the best performance is selected for the remainder of the study. The training for the tuning phase is performed using all the variables listed in Table 3 as input features to the CNNs.

The choice of hyperparameters can influence the learning phase and the final performance of a fitted model. However, in order to focus solely on the choice of the model and the impact of input size on the performance, the models are trained

with fixed hyperparameters. Stochastic gradient descent (SGD) is used with a learning rate of $\mathrm{lr} = 10^{-4}$, a weight decay of $w_d = 10^{-3}$ and a momentum of $\mathrm{m} = 0.9$. The loss function is the binary cross entropy (BCE), the training mini-batch size is



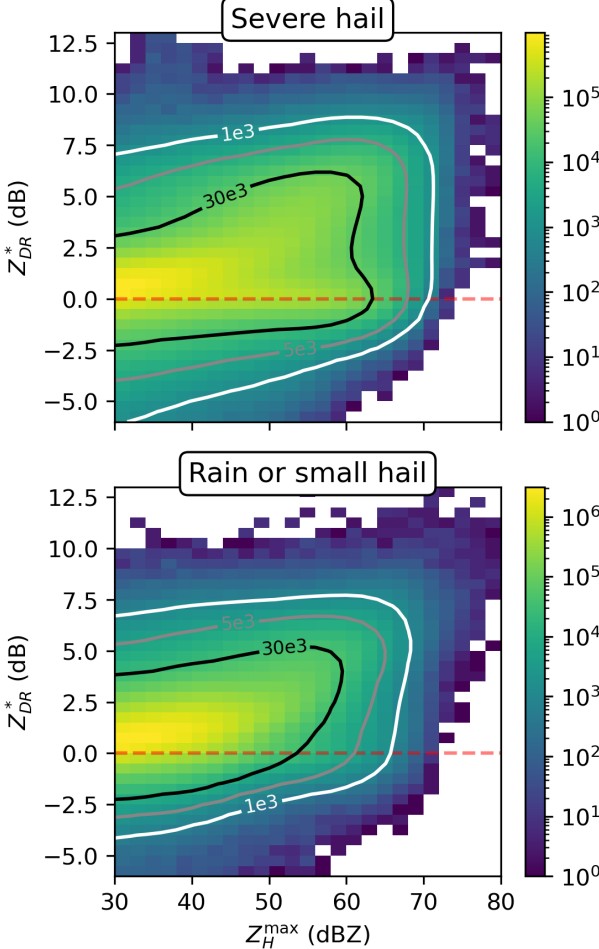

**Figure 7.** Bivariate distributions of $Z_H^{\max}$ and $Z_{DR}^*$ within $60\,\mathrm{km} \times 60\,\mathrm{km}$ images for the severe hail dataset and the rain or small hail dataset. Contours represent the frequency of values per two-dimensional bin.

bs $= 64$, and the maximum number of epochs is $n_{\mathrm{epochs}} = 300$. Additional regularisation is achieved through the incorporation of batch normalisation layers within the models. The selection of hyperparameters is highly empirical and dependent on the specific problem being solved, as well as the quality and quantity of data used for training. The aforementioned hyperparameters

are selected in order to ensure that the model's loss decreases monotonically during training towards convergence.

During the tuning phase, all possible combinations of models and input sizes are trained under identical conditions. The whole dataset containing severe hail and rain or small hail samples (7523) is separated between a training dataset, a validation dataset and a test dataset. The different splits are presented in Table 4. The training and validation datasets are employed during the tuning phase, while the test dataset is reserved for subsequent performance analysis. To ensure independence between the

datasets, samples are grouped by date. This guarantees that each date is only present in one dataset. Furthermore, an additional




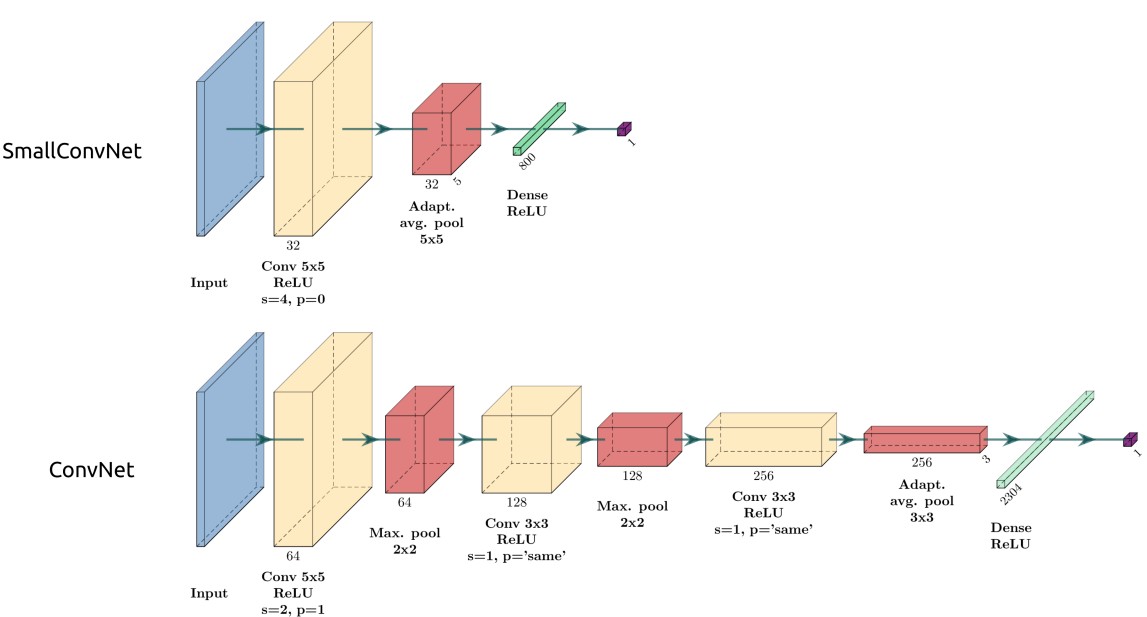

**Figure 8.** Two feed-forward CNN architectures tested in this study: the SmallConvNet and the ConvNet. Convolutional layers are denoted as 'Conv' (yellow boxes); pooling layers are denoted as 'Max. pool' and 'Adapt. avg. pool' for max pooling and adaptative average pooling respectively (red boxes); fully connected layers of perceptrons are denoted as 'Dense' (green boxes). 'p' for padding, 's' for stride. Number of filters per layer is showed below boxes. The kernel size is shown by multiplicative terms. All activation functions are ReLU. A batch normalization layer is added after each convolutional layer for regularization (hidden). The output of the network is a real number, which is subsequently passed to a sigmoid function to produce a probability of severe hail on the ground within the image, denoted as $P \in [0,1]$.

**Table 4.** Number of samples in the training, validation and test data sets for the tuning phase in section 3.2.

|  | Training | Validation | Test |
|---|---|---|---|
| Severe hail (1) | 1476 | 413 | 446 |
| Rain or small hail (0) | 3100 | 1138 | 950 |
| **Total** | **4576** (61 %) | **1551** (21 %) | **1396**(19 %) |

precaution is taken to ensure that the proportion of severe hail and rain or small hail cases remains the same in all three datasets. In order to address the imbalance of the dataset during training, the minority class (i.e. severe hail) is oversampled using weighted random sampling. This process artificially increases the number of severe hail cases seen by the CNN at each training iteration. Finally, early stopping enables the model to halt training when the validation loss fails to decrease after 20 consecutive epochs.




**Table 5.** Contingency table

|  |  | Prediction | |
|---|---|---|---|
|  |  | severe hail | rain of small hail |
| Observation | severe hail | True Positive (TP) | False Negative (FN) |
|  | rain or small hail | False Positive (FP) | True negative (TN) |

### 3.3 Scores

The performance of the models is evaluated using a scoring methodology. For the CNNs, the output provides one probability of severe hail at the ground, denoted as $P$, for each image. The image is predicted as producing severe hail ($y_{\text{pred}}^{CNN} = 1$) or rain or small hail ($y_{\text{pred}}^{CNN} = 0$) on the ground given a discrimination threshold $\alpha$:

$$y_{\text{pred}}^{CNN} = \begin{cases} 1 \text{ (severe hail)}, & \text{if } P \geq \alpha \\ 0 \text{ (rain or small hail)}, & \text{otherwise} \end{cases} \tag{7}$$

with $\alpha \in [0, 1]$.

The comparison algorithms produce either a gridded probability or a gridded hail size as output (Fig. 5). In order to facilitate comparison with the output of CNNs, it is necessary to reduce the data of hail proxies to a single value per image. Two thresholds can be used simultaneously to determine if the image is associated with severe hail on the ground: a threshold for feature values $X$, designated $\beta_X$, and a discrimination threshold for the area $A_X$ covered by the resulting field, designated $\beta_{A_X}$. If the area of pixels above $\beta_X$ exceeds $\beta_{A_X}$, the algorithm predicts severe hail on the ground within the image as follows:

$$y_{\text{pred}}^{\text{proxy}} = \begin{cases} 1 \text{ (severe hail)}, & \text{if } X \geq \beta_X \text{ and } A_X \geq \beta_{A_X} \\ 0 \text{ (rain or small hail)}, & \text{otherwise} \end{cases} \tag{8}$$

For example, if $\beta_X = 50\%$ and $\beta_{A_X} = 10\,\text{km}^2$ for POSH, the prediction for the image will be severe hail if the area of POSH above 50 % in the image exceeds $10\,\text{km}^2$. This evaluation method allows for the study of the trade-off between a threshold on the hail proxies and the area they cover, with the objective of detecting severe hail. The various feature threshold values $\beta_X$ tested in this study for the hail proxies are presented in Table 6. For A13, three different feature threshold values are employed. These are: (i) pixels with a class above or equal to the small hail class ($\beta_X \triangleq (\geq \text{SH})$), (ii) pixels with a class above or equal to the medium hail class ($\beta_X \triangleq (\geq \text{MH})$), and (iii) pixels with a class above or equal to the large hail class ($\beta_X \triangleq (\geq \text{LH})$). This approach enables the determination of the performance for different hail class as thresholds.

The performance metrics for the predictions are defined through the use of a contingency table (Table 5). The following metrics are employed in order to compute the local performance of a model: the probability of detection (POD), also known



**Table 6.** Interval of feature threshold values ($\beta_X$) tested to assess the performance of hail proxies, e.g if $\beta_X = 25\,\mathrm{mm}$ for MESH, the performance of a model where $\mathrm{MESH} \geq 25\,\mathrm{mm}$ is assessed for different areas covered by the resulting field. Increments tested along the $\beta_X$ intervals are denoted as inc.

|  | POSH | MESH |  |
|---|---|---|---|
|  | POH$_{\text{Delobbe}}$ | MESH$_{75}$ | A13 |
|  | POH$_{\text{Delobbe}}$ | MESH$_{95}$ |  |
| $\beta_X$ | [1, 85] % | [1, 60] mm | {SH, MH, LH} |
| inc. | 1 % | 1 mm |  |

as the recall, the probability of false detection (POFD), also known as the false alarm rate, the Peirce skill score (PSS) and the precision, also known as the success ratio. They are defined as follows:

$$\mathrm{POD} = \mathrm{recall} = \frac{\mathrm{TP}}{\mathrm{TP} + \mathrm{FN}} \tag{9}$$

$$\mathrm{POFD} = \frac{\mathrm{FP}}{\mathrm{TN} + \mathrm{FP}} \tag{10}$$

$$\mathrm{PSS} = \mathrm{POD} - \mathrm{POFD} \tag{11}$$

$$\mathrm{precision} = \frac{\mathrm{TP}}{\mathrm{TP} + \mathrm{FP}}. \tag{12}$$

The precision captures how often, when a model makes a positive prediction, it turns out to be correct (Kelleher et al., 2020).
The PSS shows the tradeoff between POD and POFD. The global performance of models is evaluated by calculating the receiver operating characteristic (ROC) curves and the precision-recall curves, which illustrate the trade-off between metrics at different discrimination thresholds. Each variant of the hail proxies with a given $\beta_X$ value is considered a classifier. The performance of a classifier is evaluated by calculating the metrics for each possible discrimination area ($\beta_{A_X}$). For the CNN, each point on the curves shows the local performance for a given discrimination threshold $\alpha$. For hail proxies, each point on the
curves shows the local performance for a given $\beta_X$ and a given $\beta_{A_X}$. The areas under the curve for the ROC curve (AUC-ROC) and the precision-recall curve (AUC-Pr.Re.) are computed and used as representative metrics of the global performance of a model.

## 4 Results

### 4.1 Tuning phase

The results of the tuning phase are summarised by the learning curves of the different models (Fig. 9) and the ROC and precision-recall curves, which assess the performance on the validation split (Fig. 10).

The evolution of the training loss in Fig. 9 shows a global monotonic decrease for each model and input size, implying that some information within the features is learned by the models. However, this information may be irrelevant for severe hail detection if the fitted models do not generalise well to unseen examples. Different behaviours are seen for certain input sizes





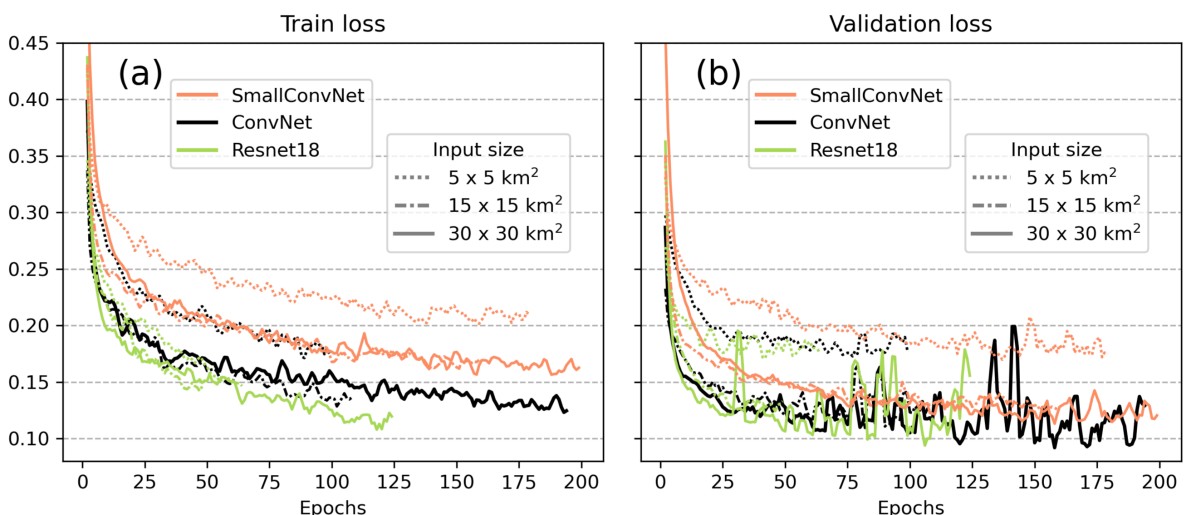

**Figure 9.** Learning curves with the evolution of the train loss **(a)** and the validation loss **(b)** for the models trained during the tuning phase. The retained model is highlighted by the solid black line. The curves are smoothed with a sliding window of 3 epochs.

and model architectures. Simple models such as the SmallConvNet lag behind in terms of minimum loss achieved on both the training and validation sets. The SmallConvNet struggles to learn as much as the other models, and reacts on average more incorrectly when presented with the validation set, especially for small input sizes (Fig. 10). This may be a classic case of underfitting, where a model is too simple to learn highly abstract features in the data. In addition to underfitting, small input sizes appear to be detrimental to the performance of CNNs, regardless of the model used. Although this was expected, it shows

that the models trained with $5\,\mathrm{km} \times 5\,\mathrm{km}$ input features are likely to miss important information in the vicinity of the storm cores that can be attributed to larger scale phenomena within the storms (hook echo, updraft region, downdraft region). The decline in performance with increasing input size is evident in Fig. 10.

Two models, the ConvNet and the ResNet18, appear to achieve equivalent performance on the validation set, despite the ResNet18 containing a significantly greater number of parameters (Fig. 9). The models in question are deeper than the Small-

ConvNet, which increases their likelihood of identifying information at varying levels of abstraction within the data, thereby enhancing their performance. The fact that the ResNet18 achieves performance levels comparable to those of the ConvNet on the validation set, despite being more complex, suggests that the size of the validation dataset may be insufficient for it to enhance its prediction. Another potential explanation is that the input sizes tested here may be too limited for ResNet architectures, which were developed for image classification on larger images (He et al., 2015).

Although a monotonic decrease is observed for the training loss across epochs, oscillations in the validation loss are evident for ConvNet and ResNet18 after the 50th epoch (Fig. 9). This behaviour is observed when a minor adjustment to the weights and biases during training results in a significant change to the value of the validation loss. This phenomenon is likely attributable





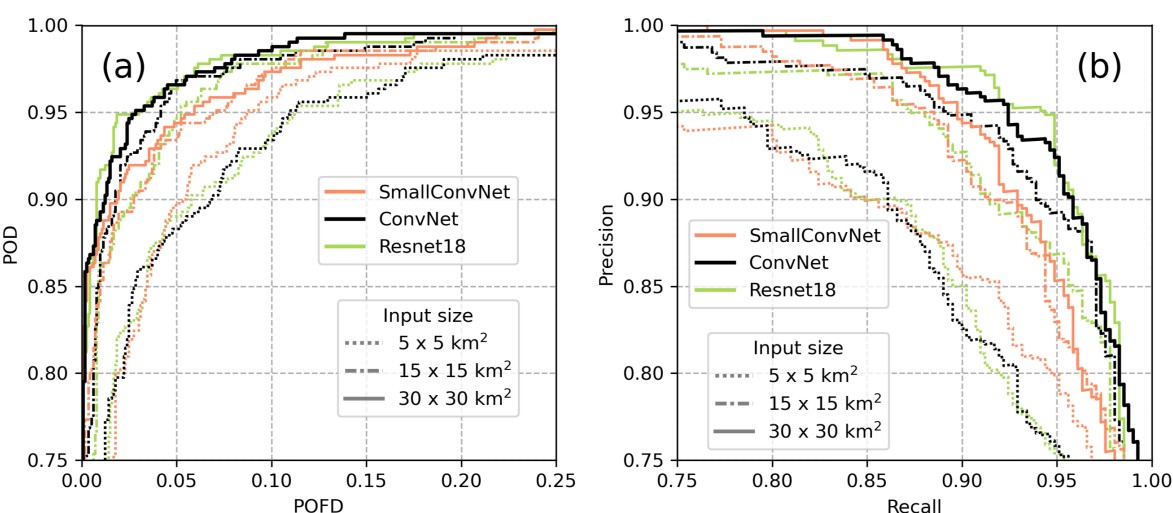

**Figure 10.** ROC curves **(a)** and Precision-Recall curves **(b)** for the models trained during the tuning phase. The retained model is highlighted by the solid black line. Models were also trained with an input size of $50\,\mathrm{km} \times 50\,\mathrm{km}$, but no amelioration was obtained (not shown).

to the relatively limited size of the validation dataset, which may prompt abrupt changes in model behaviour when parameters are updated. A direct consequence is that the models are learning additional information that may be derived from noise within the input features rather than severe hail. Although the complexity of the ConvNet and ResNet18 networks may appear to be their strength, in certain situations this may outweigh the benefits, as they are more likely to learn useless information due to their multiple layers and connections, thus overfitting. The observation that simpler models, such as SmallConvNet, do not exhibit the same degree of oscillation in the validation loss suggests that the issue may lie in the complexity of the model (Fig. 9). Nevertheless, there are methods to mitigate the adverse effects of overfitting on small datasets. One such method is cross-validation, which entails training an ensemble of models on distinct training and validation sets, and subsequently averaging the predictions of all models to obtain the final output on the test set (Kelleher et al., 2020). Models trained with an input size of $50\,\mathrm{km} \times 50\,\mathrm{km}$ were not included in the results, as they did not demonstrate any improvement in performance.

Consequently, the SmallConvNet exhibited suboptimal performance relative to deeper models, and complexity can impede generalization when utilising limited datasets. Therefore, the ConvNet with input size of $30\,\mathrm{km} \times 30\,\mathrm{km}$ is deemed an optimal compromise for the remainder of the study. Cross-validation will be employed to mitigate the risk of overfitting.

### 4.2 Feature selection and feature importance

Prior to comparing the selected CNNs with hail proxies, it is necessary to explore the features. This involves the removal of highly correlated features in order to limit them to a subset of the most useful ones and the determination of the importance of each feature in the final prediction of the CNNs.



**Figure 11.** Spearman correlation matrix for the 19 input features computed on a subset of $1 \times 10^6$ pixels from the entire dataset. Warm (resp. cold) colors correspond to positive (resp. negative) Spearman correlation coefficients.

Feature selection is performed by exploring the correlations between the 19 input features listed in Table 3. A random sample of one million pixels from the entire dataset was employed to compute the Spearman correlation coefficient between each variable. The resulting coefficient matrix is presented in Fig. 11.





It is anticipated that high positive correlations will be observed between features that are based on the same underlying variable. The MESH, $\text{MESH}_{75}$ and $\text{MESH}_{95}$ demonstrate perfect Spearman correlations (1.00) due to their underlying monotonic relationship with the SHI (see Equation (1)). The same rationale can be applied to the high positive correlations observed between ET45, $\text{POH}_{\text{Delobbe}}$ and $\text{POH}_{\text{Foote}}$, although the correlation seems higher between ET45 and $\text{POH}_{\text{Delobbe}}$ (0.98) due to its direct linear relationship with ET45 (Equation (5)). A strong positive correlation is observed between MESH variants and ET45 ($\approx 0.93$), despite the fact that they were not produced using the same methodology. The relationship between the echo tops and the integral of weighted reflectivities used in MESH may provide an explanation for this behaviour. Higher echo tops indicate a greater volume of $Z_H \geq 45\,\text{dBZ}$ above the $-20\,°\text{C}$ altitude, which carries the most weight in the construction of the SHI (Witt et al., 1998). Moderate positive correlations are observed between $Z_H^{\max}$, VIL and all the hail proxies presented in Table 5, which is consistent with expectations given their dependence on $Z_H$. The correlation between hail proxies and $\rho_{HV}$ at an altitude of $2\,\text{km}$ is moderately negative ($\approx -0.60$). This correlation is likely influenced by the effect of hail or a mixture of rain and hail on the reduction of $\rho_{HV}$ values at low levels (Kumjian, 2013a; Ryzhkov and Zrnic, 2019).

Once the correlations between variables have been established, a feature importance study can be conducted. The withdrawn variables are the following: MESH, $\text{MESH}_{95}$, $\text{POH}_{\text{Delobbe}}$ and $\text{POH}_{\text{Foote}}$. In order to prevent overfitting and to account for any potential variability in the results, the feature importance is computed by cross-validation of the performance of ten ConvNet models trained on a $30\,\text{km} \times 30\,\text{km}$ input size. A total of ten distinct combinations of training and validation sets are generated through the application of bootstrapping to the train and validation sets employed during the tuning phase (Table 4). In order to ensure the independence of the sets, the same precautions as in the tuning phase are taken. Following training, the performance of the ten fitted models is assessed on the test dataset. One variant with unperturbed input is trained for each of the ten combinations and serves as a baseline. Feature importance is then computed for each model by sequentially perturbing features using random permutations within mini-batches. If a particular feature is important to the model, its random permutation should result in decreased performance compared to the baseline model. The greater the decrease in performance, the more important the feature is for the model to detect severe hail. The performance decrease is calculated by measuring the reduction in AUC for both the ROC curve and the precision-recall curve. Fig. 12 illustrates the average and the uncertainty of feature importance for each input feature.

A low feature importance does not necessarily indicate that the feature is useless for severe hail detection. On the one hand, it may indicate that the feature plays a less important role in the output of the CNN. On the other hand, it could suggest that the majority of the information that the CNN requires to make its decision is already embedded in other features. The feature importance study only demonstrates the importance of a feature within the context of a CNN developed for severe hail detection.

The performance decline resulting from the perturbation of $\text{MESH}_{75}$ is the most pronounced among all variables. MESH was specifically developed for the detection of severe hail at S band. Consequently, despite the potential for higher reflectivity values at S band than at C band (Ryzhkov and Zrnic, 2019), it is anticipated that MESH facilitates the identification of areas with severe hail. Due to its capacity to account for the vertical reflectivity profile within the hail growth zone, MESH may be



less sensible to the effects of low vertical sampling than echo tops, and may be better at summarising information at mid- and upper-levels that are useful to quantify the severity of hail on the ground.

Three additional features appear to be important for the CNN: $Z_H^{\max}$, $\rho_{HV}^{2000}$ and ET45. This is not unexpected given that

$Z_H$ is sensitive to the particle size distribution and that high $Z_H$ values above $70\,\mathrm{dBZ}$ are typically associated with large and giant hail ($\geq 5\,\mathrm{cm}$, Ryzhkov and Zrnic, 2019). The significance of $Z_H^{\max}$ may be attributed to the finer texture of the field in comparison to 2D features extracted from the 3D grid. This may also explain the enhanced importance of $Z_H^{\max}$ relative to VIL, despite the latter having stronger correlation coefficients with hail proxies (Fig. 11). As a feature that may be negatively correlated to the presence of hail in the low levels, $\rho_{HV}^{2000}$ is of significant importance for the CNN to make its prediction. This

negative correlation of $\rho_{HV}^{2000}$ with various hail proxies indicates a decrease in $\rho_{HV}$ in the presence of hail which is expected, particularly in the presence of melting hail or hail growing in the wet regime $\rho_{HV}$ (Ryzhkov and Zrnic, 2019). Finally, it can be seen that ET45 is of some importance. Although affected by vertical sampling (Delobbe and Holleman, 2006), echo tops can contain useful information about storm height and remain relevant as a storm proxy, as more intense storms are expected to produce stronger echoes at high altitudes (Trefalt et al., 2023).

The average importance of the remaining features is situated within their respective uncertainty intervals. For instance, $Z_{DR}$ columns appear to be relatively inconsequential in the context of this study. However, this feature is not adequately represented by examining data at the time of the hailfall, as $Z_{DR}$ columns are expected to be visible prior to hailstones falling on the ground (Kuster et al., 2019). It may prove advantageous to use $Z_{DR}$ columns in the context of storm cell tracking and the study of the life cycle of storms, as it has been observed to be effective in the short-term forecast of severe weather (Kuster et al., 2019). The

relatively low importance of polarimetric collocated variables ($Z_{DR}^*$, $K_{DP}^*$, $\rho_{HV}^*$) may be explained by two factors. Firstly, as collocated polarimetric variables may originate from different heights, they may insufficiently characterize the presence of hail and important information may be lost. Secondly, this may simply reflect the fact that the value of these variables contributes little to the prediction compared to other, more significant variables such as $\mathrm{MESH}_{75}$ and $Z_H^{\max}$.

Following the completion of a feature importance study, it is standard practice to train again a model using the most important

features in order to validate its performance on unseen data. However, due to the unavailability of more severe hail reports within the French territory, it was not possible to retrain the models. Consequently, the feature importance study was limited solely to interpretation purposes.

## 4.3 Comparison with state of the art

The performance of the 10 ConvNet fitted models is compared to the hail proxies on the test set. The results are summarized

in Fig. 13 and in Table 7, Table 8 and Fig. 14.

Overall, strong AUC values are observed for all the hail proxies except A13 and POSH (Table 7). This demonstrates their capacity to optimise their performance if the threshold value above which they produce severe hail ($\beta_X$) is meticulously selected. It is in accordance with several studies that have emphasised the significance of calibration in order to optimise the performance of existing hail proxies (Murillo and Homeyer, 2019; Ortega, 2021; Brook et al., 2024; Kopp et al., 2024).





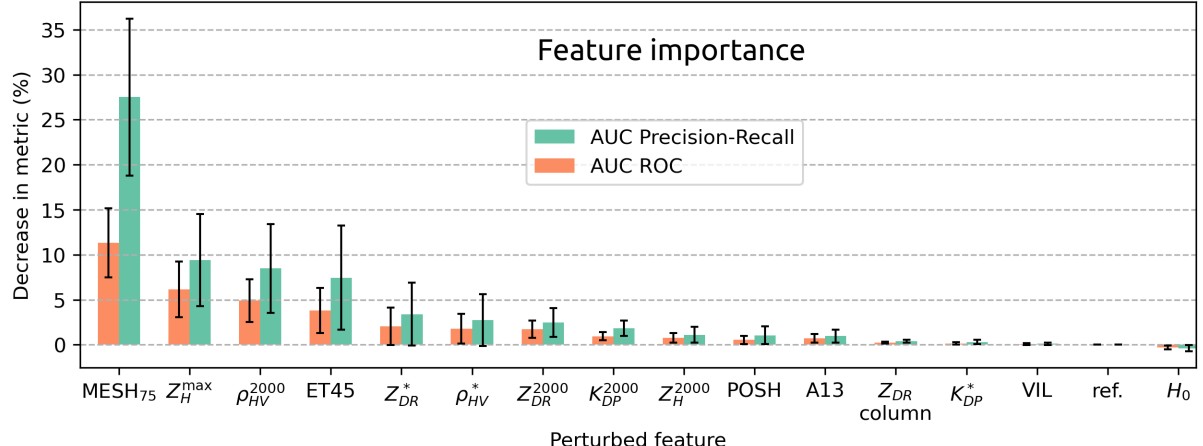

**Figure 12.** Feature importance results on the test set. Each bar corresponds to the average decrease in performance among 10 ConvNet models fitted on different combinations of training and validation sets. Uncertainty is shown as error bars of $\pm\sigma$. Ref. stands for the unperturbed model.

**Table 7.** Performance on the test set. Algorithms are compared using their five best variants producing the highest area under the ROC curve (AUC-ROC). The precision-recall AUC (AUC-Pr.Re.) and the best average threshold value is shown. Values shown as 'mean ($\pm$std)'. AUC values are multiplied by 100 for readability. $*$ represents the average metrics for three variants of A13: $\geq$ SH, $\geq$ MH and $\geq$ LH. The low average values and the wide variability for metrics on A13 are attributed to the poor performance of the $\geq$ MH and $\geq$ LH variants.

|  | AUC-ROC ($\times 100$) | AUC-Pr.Re. ($\times 100$) | $\beta_X$ |
|---|---|---|---|
| A13 | $87.30^*$ ($\pm 6.48$) | $80.92^*$ ($\pm 7.32$) | not applicable |
| $Z_H^{\mathrm{max}}$ | $92.70$ ($\pm 0.14$) | $87.55$ ($\pm 0.32$) | $55$ ($\pm 1.41$) |
| POSH | $92.82$ ($\pm 0.29$) | $90.05$ ($\pm 0.16$) | $3$ ($\pm 1.4\%$) |
| $\mathrm{POH_{Delobbe}}$ | $95.76$ ($\pm 0.05$) | $92.42$ ($\pm 0.45$) | $62$ ($\pm 5\%$) |
| $\mathrm{POH_{Foote}}$ | $95.80$ ($\pm 0.01$) | $92.35$ ($\pm 0.47$) | $26$ ($\pm 18\%$) |
| MESH | $96.31$ ($\pm 0.13$) | $92.96$ ($\pm 0.16$) | $5$ ($\pm 1.4$mm) |
| $\mathrm{MESH_{75}}$ | $96.41$ ($\pm 0.08$) | $93.10$ ($\pm 0.23$) | $20$ ($\pm 1.4$mm) |
| $\mathrm{MESH_{95}}$ | $96.45$ ($\pm 0.03$) | $93.25$ ($\pm 0.13$) | $31$ ($\pm 1.4$mm) |
| ConvNet | **$97.87$** ($\pm 0.16$) | **$96.14$** ($\pm 0.25$) | not applicable |

The validation framework developed in this study permits the further investigation of the performance of hail proxies by incorporating an additional discrimination threshold on the area covered by the feature ($\beta_{A_X}$).

     The best performance for severe hail detection overall is reached by the ConvNet model, with an average AUC-ROC of 0.979 and an average AUC-Pr-Re of 0.961 (Table 7). The low variance around mean values demonstrates a consistent behaviour among the models trained using cross-validation. Furthermore, the results indicate that the network generalises well when

EGUsphere
Author(s) 2024



applied to unseen data within the test dataset. The ConvNet exhibits the optimal trade-off between POD and POFD among all models. The number of false alarms for the best local variant of the ConvNet, i.e. the model with the highest AUC-ROC at a discrimination probability of $\alpha = 0.12$, are the lowest among all methods (61 in total), as shown in Table 8. The results demonstrate that a shallow CNN architecture is capable of identifying relevant features indicative of severe hail on the ground.

According to Table 7, the second-best algorithms for detecting severe hail on the test set are the MESH$_{95}$ and MESH$_{75}$
algorithm. Local performance in terms of PSS for the MESH$_{95}$ is the best for $\beta_X = 33\,\mathrm{mm}$ and $\beta_{A_X} = 23\,\mathrm{km}^2$. For MESH$_{75}$, the best PSS is at $\beta_X = 22\,\mathrm{mm}$ and $\beta_{A_X} = 25\,\mathrm{km}^2$. This is consistent with the findings of the feature importance study (section 4.2), which identified MESH variables as the most crucial variables for the ConvNet to detect severe hail on the ground. The feature thresholds in Table 7 are also in accordance with what can be found in other studies (Murillo and Homeyer, 2019; Ortega, 2021; Brook et al., 2024). When employed either independently or as an input feature to a CNN framework, the results
on the test set demonstrate that MESH remains effective for the discrimination of severe hail on the ground, even at C band.

The POSH and the fuzzy-logic algorithm (A13) appear to be less effective when compared to other methods, as evidenced by an AUC-ROC of 0.928 and 0.873, respectively. In the case of POSH, the application of the warning threshold (WT) in Equation (1) may be considered a potential explanation for the decrease in performance. The denser vertical sampling, higher $Z_H$ and lower attenuation of U.S. S-band radars compared to French C-band radars result in SHI values that may be smaller
than expected at S-band. Consequently, the WT fitted to the S-band radars may remove a significant proportion of pixels with low SHI values in this study. This can be verified in Fig. 5, where the POSH values cover a smaller area than other hail proxies. One potential solution would be to modify the fit of POSH in order to adapt it to the French radar network. The performance of the fuzzy-logic algorithm (A13) varies significantly depending on the hail class used as a feature threshold (i.e. $\geq$ SH, $\geq$ MH, $\geq$ LH), given the large uncertainty on the average metric values in Table 7. In essence, the performance of the algorithm
declines significantly as the threshold for hail class is increased, as the model with small hail as a threshold is the best among all other classes (Fig. 13). This may indicate a propensity of the fuzzy-logic scheme to model severe hail as small hail (SH - $< 0.5\,\mathrm{cm}$) rather than large hail (LH - $\geq 2\,\mathrm{cm}$). This may demonstrate that an improvement is possible in the design of the bi-dimensional membership functions of hail classes within A13 (see Appendix A), as the small hail and medium hail class may in reality represent larger hail sizes than those indicated.

The variation in the local performance of hail proxies for different pairs $(\beta_X, \beta_{A_X})$ is also investigated in order to demonstrate the potential for compromise in operational use. The variations in performance are presented in the form of PSS matrices in Fig. 14. The PSS matrix indicates that the local performance for a given feature threshold $(\beta_X)$ can be modified by adjusting the discrimination area $(\beta_{A_X})$. The PSS values demonstrate that the local performance of hail proxies can be markedly improved by implementing an optimised pair $(\beta_X, \beta_{A_X})$. In fact, Fig. 14 indicates that the thresholds yielding the highest PSS
for the hail proxies are not exclusive and lie within a broad range of potential feature thresholds and discrimination areas.

To investigate further the consequences of the threshold selection in terms of false alarms, two pair variants are evaluated for two of the most effective hail proxies: POH$_{\mathrm{Delobbe}}$ and MESH$_{95}$. The pairs are the following:

1. the $(\beta_X, \beta_{A_X})$ pair that produced the highest PSS among all thresholds.



2. the following pairs:

  – ($\beta_X = 50\%$, $\beta_{A_X} = 1\,\mathrm{km}^2$) for POH$_\mathrm{Delobbe}$

  – ($\beta_X = 30\,\mathrm{mm}$, $\beta_{A_X} = 1\,\mathrm{km}^2$) for MESH$_{75}$.

The latter pair variant was considered a baseline model for both proxies, where $30\,\mathrm{km} \times 30\,\mathrm{km}$ images are classified as produc-
ing severe hail if an area of at least $1\,\mathrm{km}$ is found within POH$_\mathrm{Delobbe} \geq 50\%$ and MESH$_{95} \geq 30\,\mathrm{mm}$, respectively. The results of
this local performance analysis are given as a confusion matrix in Table 8. The confusion matrix indicates a significant increase
in the number of false alarms when a small area of $1\,\mathrm{km}^2$ is used to trigger the severe hail detection for the hail proxies, in
comparison to their optimal variant. The number of false alarms increases from 68 to 364 ($+435\%$) for MESH$_{95}$ and from 80
to 552 ($+590\%$) for POH$_\mathrm{Delobbe}$. Although anticipated, the results demonstrate that incorporating fairness into the prediction of
existing hail proxies by considering both a threshold value and the area they cover is more effective than a simple verification
that would rely on the nearest hail proxy pixel within a certain radius around a location.

Additionally, the ROC curves (Fig. 13) indicate that the majority of the hail proxies compared in this study can be considered
to have equivalent skill for severe hail detection on the test set if the threshold value is chosen accordingly. This demonstrates
that the proper tuning of an operationally deployed hail detection technique can result in a satisfactory level of severe hail
detection, in accordance with other studies (Ortega, 2021; Brook et al., 2024; Ackermann et al., 2024; Kopp et al., 2024).
This interpretation as well as the threshold values may change according to the specifies of each national radar network,
particularly for different radar bands and scanning strategies where more vertical sampling is available.

        Finally, the inference of the ensemble of the ten ConvNet models is assessed on a hail event that occurred on the 11th June
2023 between 17:00 and 19:00 (UTC). The situation is extracted from the test dataset. The results are presented in Fig. 15. The
average probability of severe hail at the ground predicted by the ten models is denoted as $\overline{P}$. The computation is performed
on images with dimensions of $30\,\mathrm{km} \times 30\,\mathrm{km}$ around cell centroids every $5\,\mathrm{min}$. Cell centroids are obtained using the cell
615   identification algorithm 'tobac' (see section 2.2). Throughout the hail event and the life cycle of different cells, the ConvNet
models demonstrate a consistent behaviour. The cells responsible for the severe hail reports are accurately identified, exhibiting
a high probability of severe hail (large circle). One particular cell appears to have reached a mature stage, capable of producing
severe hail on the ground for about one hour and a half, which is consistent with the characteristics of long-lasting, highly
organised convective systems such as multicell or supercell systems. A notable proportion of cells exhibiting high reflectivity
620   ($\geq 60\,\mathrm{dBZ}$) are not identified as producing severe hail on the ground by the ConvNet models ($\overline{P} < 0.4$, grey lines without
circles). Although severe hail reports may be subject to reporting bias, this could highlight the potential of CNNs to capture
relevant information within the morphology of storms and use it to discriminate severe hail storms from other storms. The
main advantage of performing the inference with an ensemble of ConvNets is the computation of uncertainty intervals. The
uncertainty appears to increase when the predicted probability of severe hail decreases (reduced circle radius, brighter colour),
625   indicating a decline in prediction consistency when the ConvNets encounter an edge case, i.e where rain or small hail below
$2\,\mathrm{cm}$ might be produced. A small oscillation in the average probability and uncertainty is visible every $5\,\mathrm{min}$ within the north
eastern cell in Fig. 15, probably due to the different vertical sampling at each timestep implemented in the VCPs (Table 1) that



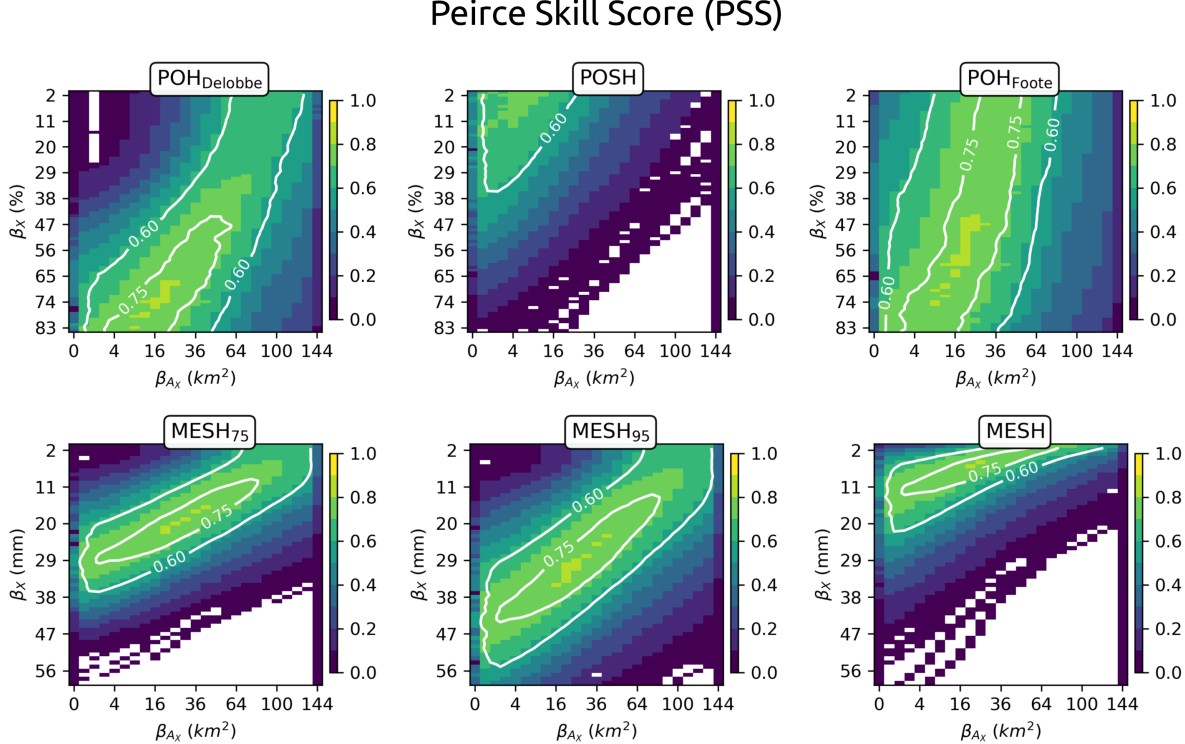

**Figure 14.** Peirce skill score (PSS) matrix for hail proxies with varying feature thresholds ($\beta_X$) and discrimination areas ($\beta_{A_X}$).

have an impact on important features of the CNN. However, a more comprehensive analysis of the inference on unseen events is necessary to gain a deeper understanding of the underlying causes of error in the prediction.

## 5 Conclusions

This study demonstrated the development and validation of a convolutional neural network (CNN) for the detection of severe hail ($\geq 2\,\mathrm{cm}$) on the ground. The framework for CNN validation, comprising a heavily filtered severe hail dataset and a rain or small hail dataset, enabled an extensive comparison of existing hail detection algorithms on the severe hail detection problem. The conclusions of this work are as follows:

1. a shallow CNN architecture, named ConvNet, was constructed and selected from among three different CNN architectures. It demonstrated superior performance for severe hail detection within radar images compared to existing hail detection algorithms on a test dataset comprising 1396 radar images with dimensions of $30\,\mathrm{km} \times 30\,\mathrm{km}$, which included severe hail and rain or small hail between 2018 and 2023. This was achieved while utilising the radar information of a unique timestep.





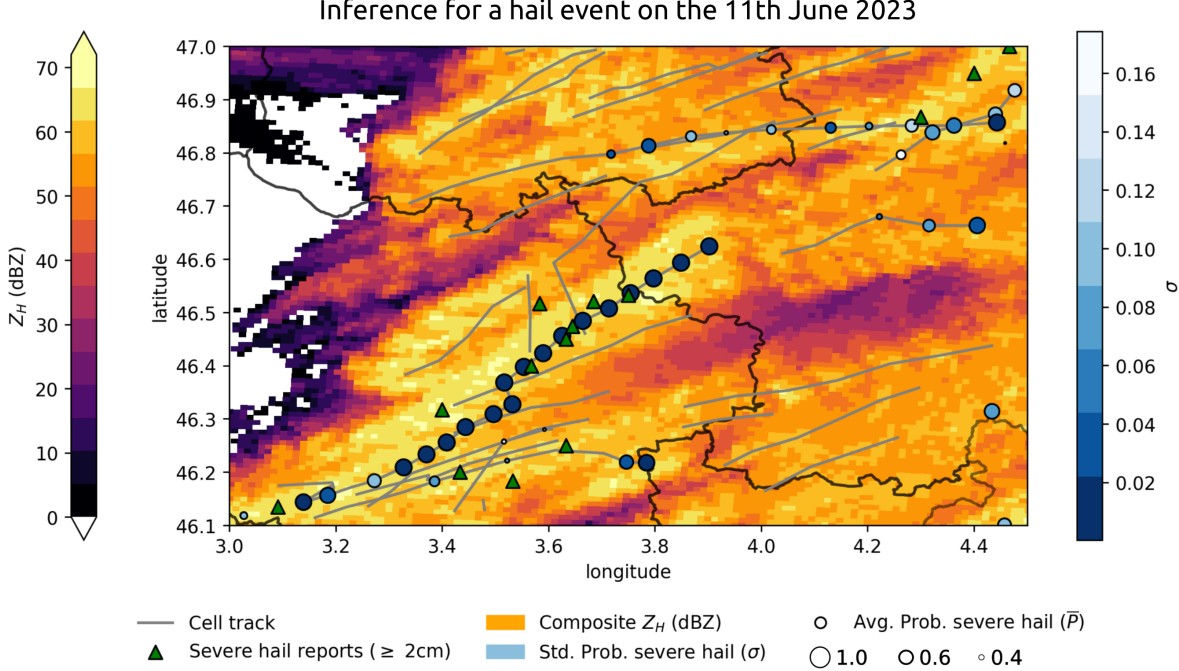

**Figure 15.** Predictions of ten ConvNet models on the 11th June 2023 between 17:00 and 19:00 (UTC). The maximum over two hours of the national reflectivity composite is shown in background (orange). Grey lines represent the cell tracks detected with the 'tobac' cell-tracking algorithm (section 2.2). Green triangles represent severe hail reports ($\geq 2\,\mathrm{cm}$) from the ESWD database within the two hours. Circles represent the cell centroids every $5\,\mathrm{min}$. Their average probability of severe hail $\overline{P}$ (circle size) and its affiliated uncertainty $\sigma$ (blue scale) are computed with the predictions of ten ConvNet models applied to $30\,\mathrm{km} \times 30\,\mathrm{km}$ images around centroids. Cell tracks without circles (pure grey lines) contain cell centroids with $\overline{P} < 0.4$.

2. a feature importance study demonstrated that incorporating hail proxies, such as MESH, as input features to the ConvNet enhanced its prediction. Other important features were $Z_H^{\max}$, ET45 and $\rho_{HV}^{2000m}$.

3. a comparison with existing hail proxies led to the conclusion that three hail proxies (MESH, POSH and POH) can be considered equivalent for severe hail detection on the test dataset if their performance is assessed using a tuned threshold value and a tuned discrimination area. Furthermore, the number of false alarms can also be drastically reduced if a threshold value and a discrimination area are chosen accordingly.

4. the study showed an example of application in real time, where the ConvNet's inference was contingent upon the detection of cell centroids via a cell identification and tracking algorithm. Its performance seemed to align with observed hail during an event within a large geographical domain. However, a more comprehensive performance validation across future events remains necessary.



The hail proxies examined in this study demonstrate satisfactory performance on the severe hail detection task when their parameters are optimised. The optimized parameters, particularly the feature threshold values $\beta_X$, align with those of previous studies (Murillo and Homeyer, 2019; Ortega, 2021). All existing hail proxies, with the exception of two, performed similarly on the test dataset. While their optimal local performance may be achieved through the use of varying threshold values and discrimination areas, it appeared that storm proxies such as echo tops for POH proxies or underlying weighted integrated

reflectivity values for MESH proxies demonstrated relevance in capturing crucial information about the presence of hail aloft. This relevance appears to be well-suited to the challenging issue of severe hail detection on the ground, based on the results of this study. The POSH exhibits suboptimal performance, likely due to the presence of a warning threshold that eliminates low SHI values at C band, owing to the low vertical sampling of French radars. The fuzzy-logic algorithm developed at Météo-France (A13), with capabilities for severe hail detection, encounters challenges due to the small and medium hail classes below

2 cm, which may represent larger hail sizes in reality.

The feature importance study yielded insights into the decision-making process of the ConvNet. The MESH proxies were identified as valuable input features, in addition to $Z_H^{\max}$, $\rho_{HV}^{2000}$ and ET45. This aligns with the strong performance of MESH proxies for severe hail detection (Table 7). The majority of the most significant variables are based on reflectivity, indicating that storm proxies based on this variable remain a valuable tool for the detection of severe hail on the ground.

One limitation of the current study is that only one timestep is used to perform a prediction associated with a report and to compare the algorithms. The life cycle of the storm is not taken into account when performing a severe hail prediction. This ultimately decreases the importance of input features that have a forecasting potential for storm severity, such as $Z_{DR}$ columns, in this study. Nevertheless, the performance of the aforementioned methods on the test set was generally satisfactory, suggesting that the reported time of the hailfall may be sufficient for the detection of severe hail in this study. However, even

after heavy filtering, uncertainty may remain regarding the location and time of severe hail. This uncertainty may compromise the generalisation of the CNN on cases that were not included in the training data, if a significant proportion of the severe hail cases on which it was trained were misplaced in space and time, or if there was a systematic error in time and location. However, this uncertainty was, as much as possible, taken into account by manually repositioning in time severe hail cases in the vicinity of a visible storm. Additionally, the construction of images of $30\,\mathrm{km} \times 30\,\mathrm{km}$ around the reports allows for a more

comprehensive view of the storms, thereby reducing the impact of potential errors in reports' location on the performance of the CNN.

The translation of the developed CNN into operations is contingent upon the implementation of a cell tracking algorithm. As the CNN was trained with radar images of storms, the storms must be identified prior to applying the CNN. The potential volatility in cell tracking due to the high sensitivity of such techniques to their input parameters can increase the inference time

of the approach, depending on the number of cells identified every five minutes. In order to detect severe hail, it is recommended to examine cells that have produced reflectivities of at least 45 dBZ. The principal advantage of this conditionality is that input features must be generated for a $30\,\mathrm{km} \times 30\,\mathrm{km}$ area centered on cell centroids, which significantly reduces the computational time required for the processing of volumetric radar data into three-dimensional grids in comparison to producing them for the entire national territory, even in areas where there is no reflectivity data that suggests the presence of hail. Furthermore,



limiting the inference to useful domains around cell centroids allows for the parallelisation of data processing and inference, which may be crucial for reducing the lag time for real-time applications.

Efforts were made to construct the input features in a way that would minimise the impact of attenuation and resolution decline with range. The use of 3D interpolated grid and volumetric radar data from the two nearest radars enabled the model to be less sensitive to these factors. However, it should be noted that extreme attenuation may not always be taken into account

in situations at the border of the French national domain. This may have an impact on the predictability of the ConvNet. The use of radar data from neighbouring countries (Germany, Switzerland, Italy, Belgium, Spain) may help to decrease the impact of attenuation in these critical regions.

Despite the implementation of precautionary measures in this study, the challenge of developing effective solutions for severe hail detection in France persists due to the scarcity of data, particularly severe hail reports. The results were analysed

on a test dataset of 1,396 radar images. While a consistent behaviour was visible in the metrics and on a broader hail event, further validation will be crucial for the CNN to validate its global performance and assess its generalisation to unseen cases. Furthermore, the specificities of the French radar network have an impact on the importance of variables and the output of the CNN in this study, particularly the radar band and the low vertical sampling. It is strongly advised that such deep learning methods be developed and tested on the specific characteristics of different national data and severe hail reports databases in

order to validate the effectiveness of CNNs in detecting severe hail on the ground. The incorporation of radar data and hail reports from neighbouring countries could significantly enhance the relevance of deep learning methods for a common hail warning system in real time.

This study establishes the foundation for the use of convolutional neural networks (CNNs) to study the morphology of storms and extract relevant information for the detection of severe hail. The interpretability of such methods is a crucial aspect.

Ongoing work includes the implementation of attribution methods that will facilitate the interpretation of the prediction of the CNN. Attribution methods for neural networks, such as saliency maps, Sobol attribution or GradCAM (Fel et al., 2022), are currently being explored in order to gain insight into the decision-making process of the CNN. Future work will probably involve the gathering of more data and the increase in the number of features, particularly polarimetric features above the melting layer. Based on the results of this study, deep learning techniques may have the potential to answer a bigger problem:

hail size estimation. Ongoing work also entails the development of a framework for the testing of such methods on the hail size estimation problem.






## Appendix A: Updated fuzzy-logic algorithm in C band from Al-Sakka et al. (2013)

The fuzzy-logic algorithm for hydrometeor classification (A13) currently operational at Météo-France corresponds to an updated version of the algorithm developed from Al-Sakka et al. (2013), with three new hail classes. The update was performed
to tackle the lack of robustness in the membership functions for hail in the original study (see conclusions of Al-Sakka et al., 2013). The following classes are now computed: 1) rain (RA), 2) wet snow (WS), 3) dry snow (DS), 4) ice (IC), 5) small hail (SH; $< 0.5\,\mathrm{cm}$), 6) medium hail (MH; $0.5\,\mathrm{cm}$ to $2\,\mathrm{cm}$) and 7) large hail (LH; $> 2\,\mathrm{cm}$). The three hail classes replace the former single hail class (HA) of Al-Sakka et al. (2013).

The fuzzy-logic scheme is based on radar variables $Z_H$, $Z_{DR}$, $\rho_{HV}$ and KDP. The brightband (BB) location is also used
and produced using the method presented by Tabary et al. (2006), which is based on the cross-correlation coefficient $\rho_{HV}$ at high elevations. Finally, the temperature $T$ is used to discriminate regions where certain hydrometeor types are not allowed. Temperature is deduced from the nearest NWP-derived sounding from the ARPEGE global model (Bouyssel et al., 2022) at the radar location.

The principle of the fuzzy-logic algorithm relies on the computation of a weight for each hydrometeor class. The hydrom-
eteor class having the highest weight becomes the hydrometeor class of the radar gate. The weight is computed thanks to membership functions (1-dimensional and 2-dimensional) built on a-priori knowledge of the single and dual-polarisation signatures for the hydrometeor classes.

The weight is defined as follows:

$$W_i^F = F^i(Z_H)F^i(T)F^i(BB)\left[F^i(Z_H, Z_{DR}) + F^i(Z_H, K_{CP}) + F^i(Z_H, \rho_{HV})\right] \tag{A1}$$

where $i$ stands for the hydrometeor type and $F$ represents the membership grade (between 0 and 1) coming from both one-dimensional and two-dimensional membership functions.

The one-dimensional membership functions $F^i(Z_H)$, $F^i(T)$ and $F^i(BB)$ for all hydrometeor types are presented in Fig. A1. As they are multiplicative terms in the weight, the presence of certain hydrometeor types is heavily driven by the reflectivity, the temperature profile at the radar site and the position of the radar gate to the BB.

The two-dimensional membership functions $F^i(Z_H, Z_{DR})$, $F^i(Z_H, K_{DP})$ and $F^i(Z_H, \rho_{HV})$ for hail depending on the relative position to the BB are presented in Fig. B1. For other hydrometeor classes, refer to Al-Sakka et al. (2013). To simplify the visualization, only regions with a membership grade superior to $0.7$ were kept, but membership grade values exist outside

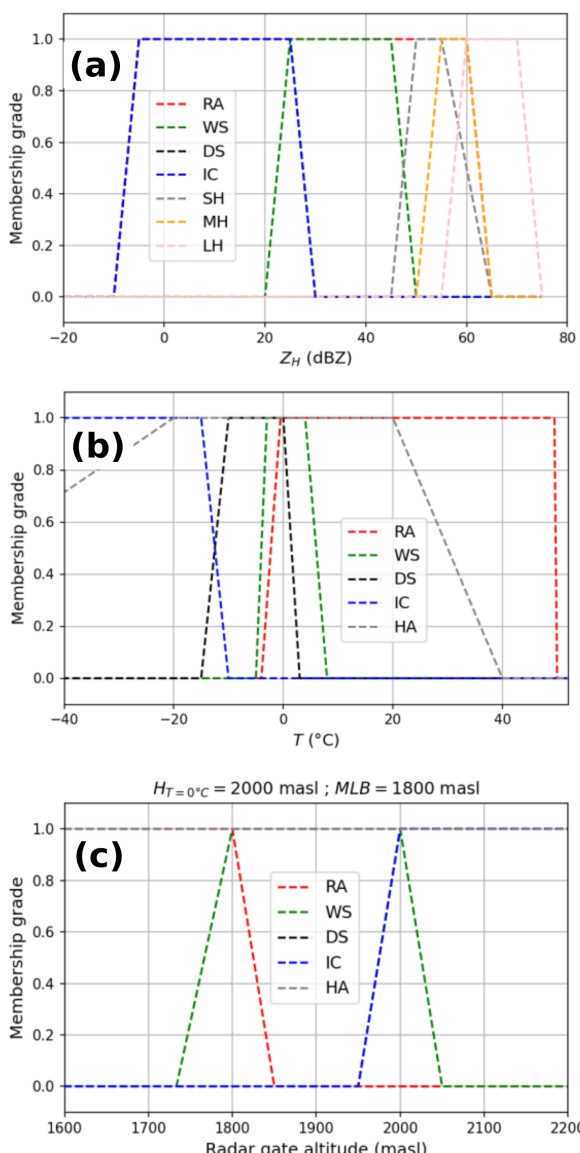

**Figure A1.** One-dimensional membership functions of the updated fuzzy-logic classification algorithm at Météo-France (A13). **(a)** $F^i(Z_H)$, **(b)** $F^i(T)$, **(c)** $F^i(\mathrm{BB})$. $F^i(\mathrm{BB})$ is shown with an altitude of freezing of $H_{T=0\,°C} = 2000$ masl (meters above sea level) computed by the AROME model, and a melting layer bottom of MLB $= 1800$ masl computed using the BB location algorithm of Tabary et al. (2006)

the intervals shown in Fig. B1.



**Figure B1.** Two-dimensional membership functions of the updated fuzzy-logic classification algorithm at Météo-France (A13) with small hail (SH), medium hail (MH) and large hail (LH). The position relative to the BB is specified as under (−), within (∼) and above (+)



*Author contributions.* VF designed the methodology, developed the code, validated the results and prepared this manuscript. CA and OC helped in the conceptualization, methodology, supervision and writing. KD contributed to the conceptualization, supervision and funding acquisition. MO contributed to the investigation of the storm mode assessment in section 2.5. CD contributed to the conceptualization of the second cell-tracking algorithm in section 2.2 and implemented the $Z_{DR}$ column detection algorithm. JF contributed to the supervision and the software of the fuzzy-logic algorithm in Appendix. OL contributed to the accessibility of the crowdsoucring reports from the Météo-France mobile application. HA contributed to the software of the fuzzy-logic algorithm in Appendix.

*Competing interests.* No competing interests.

*Acknowledgements.* This work was supported by Descartes Underwriting, Météo-France and the French national program "Les Enveloppes Fluides et l'Environnement" (LEFE, project ASMA).





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
