# Peer review of "Severe hail detection with C-band dual-polarisation radars using convolutional neural networks"

_EGUsphere, 2024_

## Referee Comment (RC1)

**Review of egusphere-2024-1336**

This is a review of "Severe hail detection with C-band dual-polarisation radars using convolutional neural networks" by Forcadell et al.

The study analyses the performance of convolutional neural networks (CNN) in detecting severe (> 2cm) from non-severe (< 2cm) hail cases. The hail cases come from different sources of observations (ESWD hail reports, ANELFA hailpad networks, MeteoFrance mobile app). Three CNN architectures are compared and trained using 19 different radar-derived features on images of four different input sizes. Features comprise existing radar-based hail proxies and polarimetric radar variables subject to ongoing research. The proxy maximum estimated size of hail (MESH) is found to be the most beneficial to the CNN as an input feature. The study also shows that existing hail proxies can be adjusted using a threshold value and a threshold area to achieve similar performance to that of the CNN for severe hail detection. Ten fitted CNN models in inference mode are used as an ensemble on hail event, allowing to compute a probability of severe hail with a related uncertainty.

This a very complete study containing several interesting results. A very limited number of studies have used CNN in the context of hail detection and the use of both existing hail proxies and polarimetric variables as input features to the CNN is innovative. The research questions are not clearly stated in the introduction. Multiple data sources are considered, and several treatments are applied to the data to identify severe and nonsevere hail cases. This makes section 2 (data and method) very dense and sometimes unclear. The choice of the CNN architecture, of the input size resolution and of the relevant features is sound and well documented. The comparison of the CNN with the "optimized" existing hail proxies not only helps assess the CNN performance in a transparent way but is also relevant for the operational use of the hail proxies themselves. The results are clearly presented and thoroughly discussed, as well as well-illustrated and summarized by good-quality figures.

The paper is highly relevant to AMT and I strongly recommend its publication, although some important revisions outlined below are required.

**General Comments**

The introduction lacks a precise description of what the authors want to investigate. Specific research questions should be listed in the introduction, and the corresponding results should be discussed with respect to those questions. This would help the authors make their point.

Section 2 should be thoroughly revised and restructured. For example, I did not fully understand the approach for constructing the rain/small hail datasets. Moreover, it contains descriptions of elements (the 2nd cell tracking algorithm used for the case study, the storm modes classification) that are not key to the paper and blur the understanding of the essential points of the methodology. Here is a list of suggestions:

- In the introduction: explain why the two datasets are needed (we have this information only at L186).
- Rename "rain or small hail" to "non-severe" for clarity (or any other shorter name)

- Use "case" instead of "report" because both reports and hailpads are used to identify severe and non-severe situations.
- Section 2.1 should be merged with 2.6 to have all information on polarimetric radar data together.
- Section 2.2: The first cell identification algorithm should be described in the section discussing the identification of non-severe hail cases. The 2nd cell identification algorithm is used only at the end of the results section for a single case study. Its description could be moved to an appendix.
- Section 2.3 is called severe-hail reports but contains a description of the three datasets used to identify both severe and non-severe cases (ESWD, ANELFA, MeteoFrance app) and is a mix between describing those datasets and how severe hail is identified.
- I suggest separating the description of the 3 datasets into a dedicated section, followed by two sections describing the identification of severe and non-severe hail cases, where the specific filtering of each dataset for this study is described. The section dedicated to non-severe hail cases would include a description of the cell tracking algorithm.
- Section 2.4 should be rewritten using a step-by-step approach.
- The categorization in storm modes made in section 2.5 is not used in the results nor further discussed. It should be moved to an appendix to improve the flow (sections 2 and 3 are already dense).

**Specific comments**

L21: "targets" – the authors could be more specific by explicitly naming the targets of interest: hydrometeors such as raindrops or hailstones.

L27: "a given content" - Do you mean an equivalent scanned volume?

L46: every detection or forecast technique will have false alarms. What level of false alarms are found for those techniques? Are they relatively high? Be more specific.

L86: in terms of hail detection as an image-based problem, the authors should consider mentioning and briefly discussing the two following references :

Gagne, D. J., Haupt, S. E., Nychka, D. W., & Thompson, G. (2019). Interpretable deep learning for spatial analysis of severe hailstorms. Monthly Weather Review, 147(8), 2827–2845. https://doi.org/10.1175/MWR-D-18-0316.1

Gagne, D. J., McGovern, A., Haupt, S. E., Sobash, R. A., Williams, J. K., & Xue, M. (2017). Storm-based probabilistic hail forecasting with machine learning applied to convection-allowing ensembles. Weather and Forecasting, 32(5), 1819–1840. https://doi.org/10.1175/ WAF-D-17-0010.1

L101: According to Fig. 1 Southeast France is covered by S and X-Band radars. Did you remove this region from the study?

L103: So this means that the time resolution of the radar variables is 15 minutes?

L104: What is the maximum distance from the radar (radius) that is considered?

L107: "the data is not corrected for advection...", but "corrected radar data was collected above hail reports". Please explain what corrections were made.

L114: What is the national reflectivity composite? Please explain briefly how it is computed.

L143: What reflectivity threshold is used here? Why use reflectivity from the nearest radar to filter ESWD reports and the Morel and Sénési (2002) cell identification algorithm to filter the Meteo France app crowdsourced report? Please explain.

L153: If hailpads reports are not used for severe hail, move this paragraph to the rain or small-hail reports (section 2.4)

L180: Is this difference in frequency a relevant factor for the study? If so, I would discuss it in more detail. If not, I would not mention it.

L184: The Meteo France crowdsourced dataset is fully described in the severe hail reports, but then it is said that it is not used to identify severe hail reports. This is confusing.

L194-L204: Why not write the filtering criteria directly in the text to avoid redundancy?

L194 + L 199: So a maximum reflectivity of at least 45 dBZ?

L199: The use of "locations" is confusing. Why not use reports (or cases)?

L204-210: This is not clear to me. The goal is to build a dataset of confirmed cases of non-severe hail that does not overlap with severe hail cases. On L203, it is said that forbidden areas are defined around all hail reports, so how are the "negative" reports outside the forbidden areas in Fig. 3 obtained? There should not be any report left. How is a hailpad with a maximum hail size > 2cm classified?

L203: 120km x 120km - I guess that you used 120km x 120km to avoid any overlap between the 60km x 60km neighborhoods around the reports. However, in section 3.2 you find that 30 km x 30 km input images contain sufficient information for the CNN. Knowing that, did you try reducing the forbidden areas to 60 km x 60 km squares to potentially increase the number of rain or small hail cases? If you do so, how many cases does this add?

L205: In the previous section, 64051 reports are mentioned. How do you get to 62854?

L209: I would distinguish the uncertainty associated with collaborative reports from the one associated with hailpads. A collaborative report can be a joke or an error and you need to filter them out (using radar reflectivity or another approach) to improve your confidence that hail indeed occurred. However, if a hailpad has multiple dents, then you are virtually certain that hail occurred; the only uncertainty you are left with is related to the time of occurrence, which is estimated by an observer.

L280: 250 m x 250 m is the horizontal resolution and 500 m the vertical?

L281: Do you have grid points that are covered by only one radar? What do you do in this case?

L282: It is not clear to me how the ROI is calculated. Can you show an example?

L295: The polarimetric data is computed for C-Band radar only. Do you only consider C-Band here as well?

Table 3: Hail proxies are named hail algorithms in section 2.7. Please use the same name throughout the paper to avoid confusion.

L328: input features: How are the features computed with respect to time? Do you use the closest radar timestep from the report time or an aggregation (maximum, average) over a time window made of several time steps? Did you test different time windows? How does it influence your results?

L348: "these features", do you mean all features or only the polarimetric ones?

L350: Is it the same interpolation that is described in section 2.6 or another one? This is not clear.

L367: How do you get from 1169 and 2605 to 2335 and 5188?

L384: Is it possible to include an illustration or a description of the ResNet architecture for comparison with those of the SmallConveNet and ConvNet? Why did you choose to compare those three CNN specifically? Can you shortly explain what are the main differences between the three CNN?

L384: input sizes: The initial image size is 60 km x 60 km. How do you get from this size to the different input sizes? (max pooling, average pooling, a window centered on the report location?)

L423: By class above or equal to small hail, do you mean the small, medium, and large classes?

L440: Please indicate the min/max value of the AUC-ROC and AUC-Pr.Re, ie. what is the best achievable performance according to those metrics (e.g.: 1)?

L445: What about the results for the 50 km x 50 km input size?

L464: "larger images": which resolution/size?

L549: Explain what each figure and table show (Fig. 13 shows the ROC and Pr.Re curves for the Conv fitted models and the hail proxies, while Table 7 shows the corresponding AUC values. Introduce Fig. 14 and Table 8 when they are discussed.

Figure 10: "Models were also trained with an input size of 50km x 50km, but no amelioration was obtained (not shown)." This should be mentioned at the beginning of the section, not in the figure caption.

L476: Move to the beginning of the section.

L485: random sample: did you limit your selection to pixels where at least one of the feature was not zero?

L526: What do you mean by significance here? Importance?

L526: "finer texture of the field": I understood that all the features had the same horizontal resolution (250m x 250m). What do you mean by texture?

Table 7: The caption misses a description of the beta_x column.

> A column with the corresponding beta_Ax values would be extremely relevant

> Several studies analyzing the skill of hail algorithms use the Critical Success Index (CSI) and the HSS (Heidke Skill Score), which can be easily computed from the contingency table. It would be interesting to have those scores in Table 7 for the best beta_x and beta_Ax pair of each score (+ those of the ConvNet).

L561: What do you mean by local here?

L565: Again, what is the meaning of local in this context?

L565: Why is the beta_x value for MESH_95 = 33 mm in the text and 31 mm in Table 7?

L578: "varies significantly depending on the hail class." Why don't you show the results for each class instead of the average? It only adds two lines to the table and the reader will have the complete information.

L606: It's not clear what accordingly refers to. You could write "...the threshold value is optimized".

L611: Do you have the predictions of MESH_95 in this case for comparison? As this is an important feature of the ConvNet, it guess that it should be similar.

Figure 14: For the two POH and POSH, why don't you show the full range of thresholds up to 100%? This is interesting information. Change the y-axis ticks labels to a multiple of 10 for readability.

L638: "utilising the radar information of a unique timestep.": This should be mentioned in the method section (see my previous remark for L328).

Figure 15: On the bottom right there are two circles without grey lines. What are they?

L646: Did you look at other hail events in detail or only this one? What were the results? This is out of curiosity and doesn't need to be included in the paper.

**Technical corrections**

L44: references should be sorted by year of publication.

L59: add "radar variables" after dual-polarization

L62: references should be sorted by year of publication.

L77: replace "traction" by attraction

L77: "In the work of Wang et al. (2018), they developed a CNN applied to..." → Wang et al. (2018) applied a CNN to...

L80: "In the work of Shi et al. (2020), they tracked..." → Shi et al. (2020) tracked...

L84: Ackermann et al. (2024) trained...

L97: "state-of-the-art"... hail detection methods?`

Table 1: I would write explicitly the 3 lowest angles on each row to avoid confusion (even if they are the same).

Figure 1 caption: in the pdf, ESWD reports appear in grey-blue instead of grey, whereas reports from the Météo-France app appear in grey instead of black.

L104: "the **three** upper elevation angles".

L106: "corrected **for** attenuation".

L153: hailpad in one word.

L223: **between**

Figure 7: The label of the grey contour is not visible, consider using another color (e.g. red). Use the same scale for both colorbars for direct comparison. The max value for rain or small hail cases is higher than for severe hail.

Table 5: True **N**egative

L457: **decreasing** instead of increasing?

L458: Define ResNet18 on L384.

L522: **sensitive** instead of sensible.

L531: remove rho_HV

L548: State of the art is subjective. Replace with an objective word, e.g.: "Comparison of CNN with **hail proxies/algorithms**".

L551: **high** instead of strong

Figure 15: the green used severe hail reports appear dark and difficult to see. You could use the same green as in Fig.3 for the reports.

---

## Referee Comment (RC3)

**Review for egusphere-2024-1336**

The study aims to analyse the performance of convolutional neural networks (CNN) in discriminating between severe (> 2cm) and non-severe (< 2cm) hail cases. Different data sources are used as hail/non-hail observations (ESWD reports, ANELFA hailpads, user reports from MeteoFrance mobile app). A comprehensive pre-selection and quality control is performed on these data to construct training, validation and test dataset. Three CNN architectures are trained using radar-derived input features on images of different input sizes. The input features comprise (polarimetric) radar data as well as radar-based hail proxies. The performance of the trained CNNs in distinguishing between severe hail and rain/small hail events is compared against hail proxies. In addition, feature selection and feature importance are discussed comprehensively. It turns out, that Maximum Estimated Size of Hail (MESH) is the most important input feature of the CNN. The CNN is able to outperform all reference proxies for different verification metrics. However, the study also shows that the discussed hail proxies are able to achieve a similar performance compared to that of the CNN, if they are adjusted/tuned regarding value threshold and area threshold.

The study is very comprehensive and contains interesting results. The results are clearly and comprehensible presented. I appreciate the wise selection and filtering of reference data for severe and non-severe hail events. The approach is well explained and discussed. Also, the CNN model architecture selection and the analysis on feature importance sounds very reasonable. I strongly recommend the publication of the study. Some minor revisions that are proposed below could further improve the paper.

**General comments**

The introduction gives a detailed overview on hail detection using remote sensing data, but the introduction on hail detection by in-situ measurements or eye-observations and the related issues (representativity, sensitivity on e.g. population density or time of day, …) is somehow missing. That's unfortunate, as these aspects are well discussed in Section 2. In addition, the specific formulation of research questions could be beneficial to outline the story of the paper.

Section 2 is quite extensive. Potentially, the discussion on storm mode (sect. 2.5) can be moved to the appendix since it does not contribute to the main storyline of the paper.

In the conclusions (sect. 5), I would appreciate a more critical discussion on the (operational) applicability of the new CNN approach also with respect to its complexity compared to the much simpler hail proxies. This is a general discussion on the costs and benefits of AI systems in Nowcasting that occurs frequently.

**Minor comments**

- ZDR-columns (Snyder et al., 2015) shall be introduced in Section 1 as a well-known precursor of hail
- The NWP perspective (i.e. environmental conditions) on hail/hail size forecasting could be shortly addressed in the introduction (e.g. Battaglioli et al., 2023)
- The details on the interpolation of polarimetric radar data on a 3D regular grid, particularly the discussion on the ROI, remains unclear. The notation "polarimetric grid" is very confusing (it's a regular grid 3D grid, not in polar coordinates?)
- Line 234: double "the"

**References**

Snyder, J. C., A. V. Ryzhkov, M. R. Kumjian, A. P. Khain, and J. Picca, 2015: A ZDR Column Detection Algorithm to Examine Convective Storm Updrafts. Wea. Forecasting, 30, 1819–1844, https://doi.org/10.1175/WAF-D-15-0068.1.

Battaglioli, Francesco & Groenemeijer, Pieter & Tsonevsky, Ivan & Púčik, Tomáš. (2023). Forecasting large hail and lightning using additive logistic regression models and the ECMWF reforecasts. Natural Hazards and Earth System Sciences. 23. 3651-3669. 10.5194/nhess-23-3651-2023.

---

## Author Comment (AC2)

**Review for egusphere-2024-1336**

**Indications of authors:** answers to each comment are in blue. Text in quotes shows portions of the modified text. Underlined text shows the modifications.

The authors would like to thank the reviewer for his/her thorough analysis of the paper that will definitely increase its quality. We hope that readers will find the answers useful.

The study aims to analyse the performance of convolutional neural networks (CNN) in discriminating between severe (> 2cm) and non-severe (< 2cm) hail cases. Different data sources are used as hail/non-hail observations (ESWD reports, ANELFA hailpads, user reports from MeteoFrance mobile app). A comprehensive pre-selection and quality control is performed on these data to construct training, validation and test dataset. Three CNN architectures are trained using radar-derived input features on images of different input sizes. The input features comprise (polarimetric) radar data as well as radar-based hail proxies. The performance of the trained CNNs in distinguishing between severe hail and rain/small hail events is compared against hail proxies. In addition, feature selection and feature importance are discussed comprehensively. It turns out, that Maximum Estimated Size of Hail (MESH) is the most important input feature of the CNN. The CNN is able to outperform all reference proxies for different verification metrics. However, the study also shows that the discussed hail proxies are able to achieve a similar performance compared to that of the CNN, if they are adjusted/tuned regarding value threshold and area threshold.

The study is very comprehensive and contains interesting results. The results are clearly and comprehensible presented. I appreciate the wise selection and filtering of reference data for severe and non-severe hail events. The approach is well explained and discussed. Also, the CNN model architecture selection and the analysis on feature importance sounds very reasonable. I strongly recommend the publication of the study. Some minor revisions that are proposed below could further improve the paper.

**General comments**

The introduction gives a detailed overview on hail detection using remote sensing data, but the introduction on hail detection by in-situ measurements or eye-observations and the related issues (representativity, sensitivity on e.g. population density or time of day, …) is somehow missing. That's unfortunate, as these aspects are well discussed in Section 2. In addition, the specific formulation of research questions could be beneficial to outline the story of the paper. For the hail detection techniques in the introduction, the authors wanted to limit themselves to radar-based hail detection techniques for the sake of simplicity. Concerning the formulation of specific scientific questions, the following paragraph has been implemented at the end of the introduction:

"[...] knowledge, none have attempted to use radar polarimetric variables for severe hail detection with CNNs. How do CNNs perform on the task of severe hail detection when applied to polarimetric radar data? Can CNNs outperform existing hail proxies? Can CNNs be used to extract information relevant to the detection of severe hail? To answer these questions, [...]"

Section 2 is quite extensive. Potentially, the discussion on storm mode (sect. 2.5) can be

moved to the appendix since it does not contribute to the main storyline of the paper.

The length argument being shared by other reviewers, the following parts have been moved to Appendix to improve the readability of the article: the description of the "second" cell identification algorithm and the storm-mode assessment.

In the conclusions (sect. 5), I would appreciate a more critical discussion on the (operational) applicability of the new CNN approach also with respect to its complexity compared to the much simpler hail proxies. This is a general discussion on the costs and benefits of AI systems in Nowcasting that occurs frequently. Details have been added in the conclusion as follows:

"[...] is recommended to examine cells that have produced reflectivities of at least 45 dBZ. The cell-identification algorithm and the production of input features for the CNN may require a greater investment of computational time and resources than existing hail proxies. The necessary 3D interpolation can be particularly costly. However, this additional computational time can be offset in real-time by the cell-identification algorithm. The input features can be generated [...]"

**Minor comments**
- ZDR-columns (Snyder et al., 2015) shall be introduced in Section 1 as a well-known precursor of hail. Section 1 (Introduction) introduces known radar-based hail detection techniques in real time. Despite being a known precursor for hail, the Zdr column, particularly the relation between their height and width with hail occurrence, is still being studied and no systematic algorithm nor extensive validation has been made so far for severe hail, to the authors' knowledge. The authors would like to limit their introduction to established hail detection techniques to remain shorter and clearer. Zdr columns and their predictive skill are anyway discussed later in the paper as they are used as an input feature.
- The NWP perspective (i.e. environmental conditions) on hail/hail size forecasting could be shortly addressed in the introduction (e.g. Battaglioli et al., 2023). The following text has been added in the introduction:

  "Other studies have employed deep learning and machine learning techniques, applied exclusively to environmental variables derived from numerical weather prediction models (NWP), for the purpose of analysing or forecasting hailstorm environments (Gagne et al. 2017; Gagne et al. 2019, Battaglioli et al. 2023)."
- The details on the interpolation of polarimetric radar data on a 3D regular grid, particularly the discussion on the ROI, remains unclear.
  The Figure below is derived from the PyArt's formulation of the ROI. The example given is computed at an altitude of z=5km. The radar's location is at (0, 0). The minimum radius is 2000m.

[Figure]

Formulas to obtain the figure are available in the Py-ART's documentation:
https://arm-doe.github.io/pyart/_modules/pyart/map/grid_mapper.html#example_roi_func_dist_beam

- The notation "polarimetric grid" is very confusing (it's a regular grid 3D grid, not in polar coordinates?). Polarimetric grid is a 3D cartesian grid with polarimetric variables (and Zh). It has been renamed as follows:

  "The Z_DR column height was calculated using the 3D Cartesian polarimetric grid"

- Line 234: double "the". It has been corrected.

**References**

Snyder, J. C., A. V. Ryzhkov, M. R. Kumjian, A. P. Khain, and J. Picca, 2015: A ZDR Column Detection Algorithm to Examine Convective Storm Updrafts. Wea. Forecasting, 30, 1819–1844, https://doi.org/10.1175/WAF-D-15-0068.1.

Battaglioli, Francesco & Groenemeijer, Pieter & Tsonevsky, Ivan & Púčik, Tomáš. (2023). Forecasting large hail and lightning using additive logistic regression models and the ECMWF reforecasts. Natural Hazards and Earth System Sciences. 23. 3651-3669. 10.5194/nhess-23-3651-2023.

---

## Author Comment (AC3)

Review comments for "Severe hail detection with C-band dual-polarisation radars using convolutional neural networks" by Forcadell et al.

Indications of authors: answers to each comment are in blue. Text in quotes shows portions of the modified text. Underlined text shows the modifications.

The authors would like to thank the reviewer for his/her thorough analysis of the paper that will definitely increase its quality. We hope that readers will find the answers useful.

This work constructed a convolutional neural network (CNN) based hail occurrence model trained on dual-polarisation radar data over France, and the training "truth" combines three ground datasets including a crow-sourcing one with careful screening and quality control. In addition to the radar measured variables, some traditional hail prediction proxies are also included as the input features. The target is to predict either severe hail (flag = 1) or rain/small hail (flag = 0). The machine learning (ML) model performance is then comprehensively compared to previously used hail detection proxies for performance statistics, and the feature importance for the model is also thoroughly evaluated. The CNN model outperforms all 6 traditional proxies in all evaluated metrics.

This work is carefully designed and thoroughly conducted. The quality-control of the training "truth" dataset involves a great amount of work, which is highly appreciated (I don't find the open science statement, but do think it would be very valuable if the training dataset can be published somewhere). The ML model architecture selection and fine-tuning are carefully executed. The results are concrete.

However, I do feel the design of the work has limited contributions to advance science, mainly because of the concern that the input features include the 6 proxies, and one of them dominates the determination processes according to the feature importance rank (and that's partially because some other highly correlated proxies are removed before feature ranking, or otherwise, they'll all rank high). So scientifically speaking, the new ML model is a "smart improved version 2.0" of the previous proxies. Since processing input features reads like not easy (e.g., interpolation using two adjacent radars to reconstruct the 3D fields, and then interpolate to 2D images, etc.), I doubt the applicability of the new ML model to operational use, given the traditional proxy data seem to be much easier to be calculated and the performance the best two traditional indices are only slightly worse (Fig. 13 and Table 8). In your revised version, I'd strongly recommend the authors adding one paragraph in the discussion or summary about their thoughts of the scientific merit and applicability of their work to the future.

Authors' comment:

It is important to note that we did not compare the "raw" hail proxies (MESH, POSH, POH) and our CNN approach, but a heavily "optimized" version of them. In Table 7, the performance of the five best models with feature thresholds (i.e. beta_x) that lead to the best performance are shown. On top of this feature threshold, an area threshold was implemented for the hail proxies to further increase their performance and add fairness to the comparison with the CNN. The study shows that, indeed, when these two thresholds are well tuned, the performance of existing hail proxies can be drastically increased to (nearly)

reach the performance of the CNN approach. Newly added scores show that the CNN beats the existing hail proxies by a certain margin (see Table 7).

On the operational use of the CNN, the necessary time to produce the input features is easily counterbalanced by the fact that the inference is performed on detected storm cells only. Hence, the inference is only performed around cell centroids rather than the whole radar domain, drastically decreasing the processing time of volumetric radar data.

A discussion has been added in the Conclusion:

"[...] is recommended to examine cells that have produced reflectivities of at least 45 dBZ. The cell-identification algorithm and the production of input features for the CNN may require a greater investment of computational time and resources than existing hail proxies. The necessary 3D interpolation can be particularly costly. However, this additional computational time can be offset in real-time by the cell-identification algorithm. The input features can be generated [...]"

Another minor concern is the length. It's a bit too lengthy right now, easily causing readers missing highlights of your work. I'd suggest moving some definition of the common ML terminologies (ROC, AUC, confusion matrix) to the appendix, as well as detailed procedures of the QC of your training "truth". This comment being shared with another reviewer, the following parts have been moved to Appendix to improve the readability of the article: the description of the second cell identification algorithm and the storm-mode assessment.

Minor caveats:

1. Reconstructed 2D images from one closest radar and a second-closest radar are both used. But it was never discussed (or I might have missed) what are the differences for the prediction? Is it more practically to just use the closed image? Or does the result really sensitive to the distance between radar location and event location? Only the "Polarimetry" features in Table 3 are produced independently for each case using each of the two nearest radars. The other features are extracted from the 3D grid, which is created using both radars. The addition of "Polarimetry" features of the second-nearest radar was a way to augment the dataset and to make a prediction in operational use even if data missed from the first radar. It was also a way for the CNN to be less sensitive to the distance to the radar as situations where the radar is far away (by more than 130km) are used for the training, mainly coming from the second-nearest radar. Differences in the prediction were not particularly explored depending on the radar used for the "Polarimetry" features. It is expected that the predictions from the second radar using the CNN might be less robust than using the nearest radar in most of the cases, particularly if the storm is already well sampled by the first radar, or if artefacts prevent a correct sampling of the storm by the second radar (PBB, attenuation). Fig. 15 shows an inference example with the "Polarimetry" features produced with the first radar only, because it is indeed more practical. Ideally, two samples for each centroid should be created and the prediction averaged among both: one with the "Polarimetry" features from the first radar, and another with the "Polarimetry" features from the second radar.

2. Starting from comparing ML results to various previous hail detection proxy variables (Section 4.2), the ResNet is dropped for discussion. Why? Is it because training ResNet takes significantly longer time than training a ConvNet? All CNNs tested in this study took relatively the same time to train, mainly because the dataset can be considered "small". If the dataset was 10 times bigger, the question of the resources needed to train the ResNet would have arisen. The ResNet is dropped from the discussion during the tuning phase (see section 4.1). It is dropped because it reaches the same performance on the validation dataset compared to other models, but at the cost of being much more complex. More complex models on small datasets have a tendency to overfit as they are exposed to learning noise in the data rather than important features. It is shown in the evolution of the validation loss during consecutive epochs (Fig. 09), where huge oscillations are rapidly visible for the ResNet.

3. How was "SHI" defined? It was never clear to me. If it has an explicit analytical format involving radar measured quantities, then it's a "smart use" of radar measurements, and you can directly use "SHI" as your training input feature.The definition of the SHI has been added to the text in the form of Equations in section 2.5:

$$\text{SHI} = 0.1 \int_{H_0}^{H_t} W_T(H_T)\dot{E}dH,$$

with

$$\dot{E} = 5 \times 10^{-6} \times 10^{0.084Z} W(z),$$

$$W_T(H) = \begin{cases} 0 & \text{for } H \le H_0 \\ \dfrac{H - H_0}{H_{-20} - H_0} & \text{for } H_0 < H \le H_{-20}, \\ 1 & \text{for } H \ge H_{-20} \end{cases}$$

$$W(Z) = \begin{cases} 0 & \text{for } Z_H \le Z_L \\ \dfrac{Z_H - Z_L}{Z_U - Z_L} & \text{for } Z_L < Z_H < Z_U, \\ 1 & \text{for } Z_H \ge Z_U \end{cases}$$

The SHI could have been added as an input feature as well, while removing its highly correlated counterparts. It is anyway indirectly implemented in MESH proxies and would be perfectly Spearman correlated to MESH. The importance of SHI in the input can be seen as the importance of MESH, and vice versa.

---

## Author Comment (AC4)

**Review of egusphere-2024-1336**

**Indications of authors:** answers to each comment are in blue. Text in quotes shows portions of the modified text. Underlined text shows the modifications.

The authors would like to thank the reviewer for his/her thorough analysis of the paper that will definitely increase its quality. We hope that readers will find the answers useful.

This is a review of "Severe hail detection with C-band dual-polarisation radars using convolutional neural networks" by Forcadell et al.

The study analyses the performance of convolutional neural networks (CNN) in detecting severe (> 2cm) from non-severe (< 2cm) hail cases. The hail cases come from different sources of observations (ESWD hail reports, ANELFA hailpad networks, MeteoFrance mobile app). Three CNN architectures are compared and trained using 19 different radar-derived features on images of four different input sizes. Features comprise existing radar-based hail proxies and polarimetric radar variables subject to ongoing research. The proxy maximum estimated size of hail (MESH) is found to be the most beneficial to the CNN as an input feature. The study also shows that existing hail proxies can be adjusted using a threshold value and a threshold area to achieve similar performance to that of the CNN for severe hail detection. Ten fitted CNN models in inference mode are used as an ensemble on hail event, allowing to compute a probability of severe hail with a related uncertainty.

This a very complete study containing several interesting results. A very limited number of studies have used CNN in the context of hail detection and the use of both existing hail proxies and polarimetric variables as input features to the CNN is innovative. The research questions are not clearly stated in the introduction. Multiple data sources are considered, and several treatments are applied to the data to identify severe and nonsevere hail cases. This makes section 2 (data and method) very dense and sometimes unclear. The choice of the CNN architecture, of the input size resolution and of the relevant features is sound and well documented. The comparison of the CNN with the "optimized" existing hail proxies not only helps assess the CNN performance in a transparent way but is also relevant for the operational use of the hail proxies themselves. The results are clearly presented and thoroughly discussed, as well as well-illustrated and summarized by good-quality figures. the paper is highly relevant to AMT and I strongly recommend its publication, although some important revisions outlined below are required.

**General Comments**

The introduction lacks a precise description of what the authors want to investigate. Specific research questions should be listed in the introduction, and the corresponding results should be discussed with respect to those questions. This would help the authors make their point. Section 2 should be thoroughly revised and restructured. For example, I did not fully understand the approach for constructing the rain/small hail datasets. Moreover, it contains descriptions of elements (the 2nd cell tracking algorithm used for the case study, the storm modes classification) that are not key to the paper and blur the understanding of the essential points of the methodology. Here is a list of suggestions:

- In the introduction: explain why the two datasets are needed (we have this information only at L186). A mention of the two datasets has been added to the abstract and an explanation has been added at the end of the Introduction:

"The framework developed herein for the detection of severe hail on the ground comprises the training of CNNs to discriminate between severe hail cases (>= 2cm) and rain or small hail cases (< 2cm). To this end, a dataset comprising both types of cases is constructed."

- Rename "rain or small hail" to "non-severe" for clarity (or any other shorter name). "Non-severe" was thought misleading as the negative dataset (i.e. the rain or small hail dataset) may contain non-severe hail and rain cases. If it were called the "non-severe" dataset, readers would think it may include only non-severe hail (i.e. hail between 5mm and 20mm), which is not the case here as it also probably includes rain-only instances. As a result, we keep the "rain or small hail" for clarity.
- Use "case" instead of "report" because both reports and hailpads are used to identify severe and non-severe situations. Changed to "case" when appropriate throughout the article. A few examples are shown here and in section 2.4:

  "[...]comprises the training of CNNs to discriminate between severe hail cases (>= 2cm) and rain or small hail cases ( < 2cm) [...]"

  "[...]  Volumetric radar data is not corrected for advection between successive elevation angles. Radar data was collected for severe hail cases (see section 2.3) and for rain or small hail cases (see section 2.4)."

  "[...] these reports were not employed in the creation of severe hail cases [...]"

  "Rain or small hail cases are created as situations that produced either rain or small hail below 2cm."

- Section 2.1 should be merged with 2.6 to have all information on polarimetric radar data together. Section 2.6 has been moved at the end of section 2.1 with the following transition sentence.:

  "In addition to the processed polarimetric radar variables available in the polar radar geometry, three-dimensional grids are generated for the study."

- Section 2.2: The first cell identification algorithm should be described in the section discussing the identification of non-severe hail cases. Only the "first" cell identification algorithm, i.e. the one that assists the creation of Rain or small hail cases, is now described in the text. Section 2.2 has been removed from the text and the description has been moved in a newly created section 2.4 called "Rain or small hail cases". The 2nd cell identification algorithm is used only at the end of the results section for a single case study. Its description could be moved to an appendix. The 2nd cell identification algorithm (for inference) has been moved to Appendix A.
- Section 2.3 is called severe-hail reports but contains a description of the three datasets used to identify both severe and non-severe cases (ESWD, ANELFA, MeteoFrance app) and is a mix between describing those datasets and how severe hail is identified. See next comment.
- I suggest separating the description of the 3 datasets into a dedicated section, followed by two sections describing the identification of severe and non-severe hail cases, where the specific filtering of each dataset for this study is described. The

section dedicated to non-severe hail cases would include a description of the cell tracking algorithm. The description of the hail reports databases, severe hail cases and rain or small hail cases has been reorganised as follows:

- The "second" cell identification algorithm for inference has been moved to Appendix A.
- Section 2.2 now presents the "Hail reports" databases
- Section 2.3 now presents the "Severe hail cases" and the processing of the ESWD reports.
- Section 2.4 ("Rain or small hail cases") now presents the cell-identification algorithm, the filtering of the collaborative reports, the step-by-step approach to pick time and locations for the "Rain or small hail cases", and the last filtering to remove mild precipitation cases from the dataset.

- Section 2.4 should be rewritten using a step-by-step approach. A paragraph has been added at the beginning of the section:

  "[...]The creation of rain or small hail cases is divided into four distinct phases. The first phase involves the presentation of the cell-identification algorithm. The second phase entails the implementation of a consistency check to filter the collaborative reports using the cell-identification algorithm. The third phase encompasses the successive steps to identify the time and locations of the rain or small hail cases. The final phase comprises a filter to exclude mild precipitation cases from the dataset.[...]"

  Then, each paragraph is introduced by its order of appearance: "Firstly", "Secondly", "Thirdly" and "Fourthly".

- The categorization in storm modes made in section 2.5 is not used in the results nor further discussed. It should be moved to an appendix to improve the flow (sections 2 and 3 are already dense). The "second" cell-identification algorithm has been moved to Appendix B.

**Specific comments**

L21: "targets" – the authors could be more specific by explicitly naming the targets of interest: hydrometeors such as raindrops or hailstones. This suggestion has been implemented:

"The echoes returned from targets such as raindrops or hailstones [...]"

L27: "a given content" - Do you mean an equivalent scanned volume? We mean a mass per volume of air. The sentence has been rewritten for clarity:

"For a given amount of hail contained in a unit volume of cloud, i.e. a given hail content, the hail size distribution is shifted towards larger diameters in comparison to rain. This results in higher reflectivities for hail compared to rain"

L46: every detection or forecast technique will have false alarms. What level of false alarms are found for those techniques? Are they relatively high? Be more specific. « Relatively high amount» has been added and CSI values were extracted from the references as follows:

"While providing a high probability of detection depending on the validation methodology, these techniques are known to suffer from a relatively high amount of false alarms and moderate critical success indices (CSI between 0.4 and 0.6, Holleman, 2001; Ortega, 2021; Pilorz et al., 2022)

To get more details, the authors invite the readers to take a look at the references at the end of the line.

L86: in terms of hail detection as an image-based problem, the authors should consider mentioning and briefly discussing the two following references

Gagne, D. J., Haupt, S. E., Nychka, D. W., & Thompson, G. (2019). Interpretable deep learning for spatial analysis of severe hailstorms. Monthly Weather Review, 147(8), 2827–2845.

https://doi.org/10.1175/MWR-D-18-0316.1

Gagne, D. J., McGovern, A., Haupt, S. E., Sobash, R. A., Williams, J. K., & Xue, M. (2017). Storm-based probabilistic hail forecasting with machine learning applied to convection-allowing ensembles. Weather and Forecasting, 32(5), 1819–1840. https://doi.org/10.1175/ WAF-D-17-

0010.1 -

The following has been added:

"Other studies have employed deep learning and machine learning techniques, applied exclusively to environmental variables derived from numerical weather prediction models (NWP), for the purpose of analysing or forecasting hailstorm environments (Gagne et al. 2017; Gagne et al. 2019, Battaglioli et al. 2023)."

L101: According to Fig. 1 Southeast France is covered by S and X-Band radars. Did you remove this region from the study? X-band radars were not used in the study. Then, regions where the two nearest radars were not C-band radars were withdrawn as well. Explanation has been added in the text:

"This study uses data from C-band radars within metropolitan France (Fig. 1). It did not include S- nor X-band radars. Only cases where the two nearest radars were C-band radars were considered in this study."

L103: So this means that the time resolution of the radar variables is 15 minutes? No, it is 5min. Three different 5min cycles are included in a 15 minutes super-cycle. It is mentioned in Table 1.

L104: What is the maximum distance from the radar (radius) that is considered? The maximum distance from the radar is 250km for all the C-band radars of Météo-France. It has been implemented in the text:

"The maximum range of the radars is 250 km.."

L107: "the data is not corrected for advection…", but "corrected radar data was collected above hail reports". Please explain what corrections were made. The second « corrected » was removed from the text to avoid confusion. The data is corrected from attenuation, partial beam blockage and non-meteorological echoes, but not from advection:

"The raw volumetric radar data, with a range resolution of 240m and an azimuthal sampling of 0.5°, are processed through a polarimetric processing [...]. Non-meteorological echoes are removed, partial beam blockage is corrected, and Z_H and Z_DR are corrected for attenuation [...]. Volumetric radar data is not corrected for advection between successive elevation angles"

L114: What is the national reflectivity composite? Please explain briefly how it is computed. It is deduced from the lowest available and valid reflectivity measurement from all the radars. We encourage readers to follow the reference mentioned at the end of the line for more information. Explanation has been added in the text:

"and subsequently applied to the national reflectivity composite product, whereby the lowest available and valid reflectivity measurement from all the radars is selected (Caumont et al. 2021)"

L143: What reflectivity threshold is used here? Why use reflectivity from the nearest radar to filter ESWD reports and the Morel and Sénési (2002) cell identification algorithm to filter the Meteo France app crowdsourced report? Please explain.

The ESWD reports were the first reports to be gathered in the timeline of the study, and were considered of much better quality compared to the Meteo-France app crowdsourced reports. Because of that, only a visual consistency check was performed on the ESWD reports to check if a cell was visible around the time mentioned in the report at the report location. No hard reflectivity threshold was used, but we were looking for reflectivities above 45dBZ at different elevations. It was a way to discard evident null cases rather than heavily filter the database. At that time of the study, the reflectivity field of the nearest radar was the easiest way to perform such a test as it was already available locally.

For the Meteo-France crowdsourced reports, given their quantity and the number of days they covered, checking the nearest volumetric radar data was not deemed possible, as a lot of wrong hail reports exist in the database and getting corrected volumetric radar data for every report would have been extremely timely expensive. It was more appropriate to download and use light cell objects that were already identified and archived at Météo-France to perform the consistency check on the crowdsourced reports.

L153: If hailpads reports are not used for severe hail, move this paragraph to the rain or small-hail reports (section 2.4). See General comments.

L180: Is this difference in frequency a relevant factor for the study? If so, I would discuss it in more detail. If not, I would not mention it. It was removed from the text.

L184: The Meteo France crowdsourced dataset is fully described in the severe hail reports, but then it is said that it is not used to identify severe hail reports. This is confusing. It has been moved to the newly created "Rain or small hail cases" section to avoid confusion. See General comments above.

L194-L204: Why not write the filtering criteria directly in the text to avoid redundancy? The text from L193 to L198 has been shortened.

BEFORE: "Several precautions were taken to build this database. First, times and locations with no potential for hail formation were excluded based on a minimum reflectivity threshold. Thus, a disproportionate number of useless cases to train the CNN were discarded. Second, locations in sparsely populated areas and times of day when hail cannot be reported were excluded, as was done in the study by Kopp et al. (2024). Finally, entire areas during time intervals around hail reports were forbidden to avoid domains where severe hail was highly probable. As a result, an initial filtering was applied every 20 min using cell objects"

AFTER: "A number of measures were implemented to prevent the inclusion of irrelevant cases where hail was deemed unlikely and to ensure the integrity of the rain or small hail database, which shall not include severe hail cases."

L194 + L 199: So a maximum reflectivity of at least 45 dBZ? Yes.

L199: The use of "locations" is confusing. Why not use reports (or cases)? "Locations" was chosen instead of "cases" to make sure the reader understood that we were looking for a point location in time under the following criteria (mentioned in the text). Despite "cases" being convenient, its meaning would differ in other places of the text.

L204-210: This is not clear to me. The goal is to build a dataset of confirmed cases of non-severe hail that does not overlap with severe hail cases. On L203, it is said that forbidden areas are defined around all hail reports, so how are the "negative" reports outside the forbidden areas in Fig. 3 obtained? There should not be any report left. How is a hailpad with a maximum hail size > 2cm classified?. The goal is to build a dataset of confirmed rain or small hail cases (see new section 2.4). They are not pre-existing in any database. Possible candidates to populate the rain or small hail database are candidates outside forbidden areas (60km x 60km around known hail reports) that meet the list of criteria in section 2.4 (high enough reflectivity, minimal population density, …). A hailpad with a maximum hail size above 2cm is not classified, it is just used to create a "forbidden" area.

L203: 120km x 120km - I guess that you used 120km x 120km to avoid any overlap between the 60km x 60km neighborhoods around the reports. However, in section 3.2 you find that 30 km x 30 km input images contain sufficient information for the CNN. Knowing that, did you try reducing the forbidden areas to 60 km x 60 km squares to potentially increase the number of rain or small hail cases? If you do so, how many cases does this add? We did not. We expect the number of cases to increase, but not linearly with the reduction in area, as the criteria on population density is the most restrictive to pick a location for a rain or small hail case.

L205: In the previous section, 64051 reports are mentioned. How do you get to 62854? 62854 is the right number, as shown in Fig. 1. It has been modified accordingly.

L209: I would distinguish the uncertainty associated with collaborative reports from the one associated with hailpads. A collaborative report can be a joke or an error and you need to filter them out (using radar reflectivity or another approach) to improve your confidence that hail indeed occurred. However, if a hailpad has multiple dents, then you are virtually certain that hail occurred; the only uncertainty you are left with is related to the time of occurrence, which is estimated by an observer. Text has been modified to separate both uncertainties:

"Using a filter that combines all available hail reports to exclude 'forbidden' areas where rain or small hail cases cannot be created was considered the best option, given the significant uncertainty in the size and hailfall time in the hailpad measurements and in the overall robustness of the collaborative reports."

L280: 250 m x 250 m is the horizontal resolution and 500 m the vertical? Yes.

L281: Do you have grid points that are covered by only one radar? What do you do in this case? Given the dense network of French radars, the low topography of hail risky areas in France, and the consideration of only C-band regions, only a very small portion of cases may not be covered by two C-band radars. In case a 3D grid is constructed from one radar only, we (purposely) do nothing to fit as much as possible operational conditions.

L282: It is not clear to me how the ROI is calculated. Can you show an example? The Figure below is derived from the PyART's formulation of the ROI. The example given is computed at an altitude of z=5km. The radar's location is at (0, 0). The minimum radius is 2000m.

[Figure]

Formulas to obtain the figure are available in the Py-ART's documentation:
https://arm-doe.github.io/pyart/_modules/pyart/map/grid_mapper.html#example_roi_func_dist_beam

L295: The polarimetric data is computed for C-Band radar only. Do you only consider C-Band here as well? Yes. It was just to mention that the operational fuzzy-logic algorithm also has capabilities for S and X-band.

Table 3: Hail proxies are named hail algorithms in section 2.7. Please use the same name throughout the paper to avoid confusion. "Algorithm" was replaced by "hail proxy" when appropriate throughout the paper.

L328: input features: How are the features computed with respect to time? Do you use the closest radar timestep from the report time or an aggregation (maximum, average) over a time window made of several time steps? Did you test different time windows? How does it influence your results? Input features are created at the time mentioned in the cases. They include radar data at the closest radar timestep from the reported time. French radars are synchronised. No time windows were tested nor used. It has been clarified in the text:

"The input features are summarised in Table . They are produced using the nearest radar timestep from the time mentioned in each case."

L348: "these features", do you mean all features or only the polarimetric ones? We mean the original volumetric fields (Zh, Zdr, Kdp, Rhohv) before 3D interpolation. It has been corrected as follows:

"The utilisation of 3D interpolation may result in the loss of information present in the original volumetric fields, as it reduces the small scale variations and the original resolution of the fields."

L350: Is it the same interpolation that is described in section 2.6 or another one? This is not clear. It is a different one. It has been clarified in the text:

"[...] in order to match the horizontal resolution of the 3D grid (250 m x 250 m). This interpolation is different from the 3D interpolation scheme in section 2.1."

L367: How do you get from 1169 and 2605 to 2335 and 5188? Two samples / radar images were created for each case : one with the nearest radar and another with the second nearest radar. Then, the cases were visually checked again and a small number were manually removed due to interpolation issues. (1169 x 2 radars – 3) + (2605 x 2 radars – 22) = 2335 + 5188 = 7523. Precision has been added to the text:

"A total of 7523 radar samples were produced. Among them, 2335 were created from the 1169 severe hail cases, and 5188 were created from the rain or small hail cases. A total of 3 severe hail samples and 22 rain or small hail samples were removed from the dataset due to issues with interpolation, primarily arising from the second-nearest radar."

L384: Is it possible to include an illustration or a description of the ResNet architecture for comparison with those of the SmallConveNet and ConvNet? As the article is already quite lengthy, a figure of a ResNet would make the article even heavier. We encourage readers to check the reference in the text if they are looking for visuals (He et al., 2015). Why did you choose to compare those three CNN specifically? Both SmallConvNet and ConvNet were empirically created for the occasion to test shallow CNN architectures inspired from the AlexNet architecture (Krizhevsky et al., 2017) on the problem of severe hail detection. Two of them were proposed with different complexities (i.e. depths) to study the impact of network complexity on the predictions. For the ResNet, as it is state-of-the-art in classification of image recognition problems, it was deemed important to add it to the list. It was also a way to show that, while tempting, using really deep architectures like the ResNet on the problem of severe hail detection is not the best way to go as complex models are more prone to overfitting on small datasets. Can you shortly explain what are the main differences between the three CNN? SmallConvNet et ConvNet are feed-forward CNNs. Information only goes from left to right. ResNet includes skipped connections where the input of a layer is added to its output. ResNet18 is a variant of the ResNet that is (much) deeper than the SmallConvNet and the ConvNet, with many more convolutional layers. We invite the readers to check the reference for more information about ResNets (He et al., 2015).

L384: input sizes: The initial image size is 60 km x 60 km. How do you get from this size to the different input sizes? (max pooling, average pooling, a window centered on the report location?) A window centered on the case location. It has been added to the text:

"Four input sizes are tested with the different models using a centered crop around the case location."

L423: By class above or equal to small hail, do you mean the small, medium, and large classes? Yes.

L440: Please indicate the min/max value of the AUC-ROC and AUC-Pr.Re, ie. what is the best achievable performance according to those metrics (e.g.: 1)? 1.0: all predictions are correct, 0.0: all predictions are incorrect. Added in the text:

"If all the predictions are wrong (resp. right), the AUC is 0.0 (resp. 1.0). In the context of a balanced dataset, an AUC of 0.5 indicates that the model's performance is equivalent to that of a random function."

L445: What about the results for the 50 km x 50 km input size? See comment Figure 10 and L476.

L464: "larger images": which resolution/size? ResNet architectures were trained and validated on images from 32 x 32 pixels (CIFAR dataset) to much higher resolutions (256 x 256 commonly used for the ImageNet dataset). See He et al. (2015). The statement L464 has been removed from the text as it was incorrect.

L549: Explain what each figure and table show (Fig. 13 shows the ROC and Pr.Re curves for the Conv fitted models and the hail proxies, while Table 7 shows the corresponding AUC values. Introduce Fig. 14 and Table 8 when they are discussed. Fig. 14 and Tab. 8 description have been moved where they are introduced in the text. Explanation of Fig. 13 and Table 7 has been added to the text at the beginning of the section as follows:

"The performance of the 10 ConvNet fitted models is compared to the hail proxies on the test set. The results are summarized in Fig. 13 as ROC and Precision-Recall curves. Table 7 summarizes the global metrics with the feature threshold values leading to the best performance."

Figure 10: "Models were also trained with an input size of 50km x 50km, but no amelioration was obtained (not shown)." This should be mentioned at the beginning of the section, not in the figure caption. Deleted + see comment L476.

L476: Move to the beginning of the section. Done

L485: random sample: did you limit your selection to pixels where at least one of the feature was not zero? No. However, as the Spearman correlation coefficient is based on the rank of the variables, zeros do not influence the final result.

L526: What do you mean by significance here? Importance? We mean importance. It has been changed to importance.

L526: "finer texture of the field": I understood that all the features had the same horizontal resolution (250m x 250m). What do you mean by texture? By texture, we mean small scale variation of the fields. Texture has been removed from the text to avoid confusion, and replaced by "small scale variations". See previous comment L346

The difference in "small scale variations" between collocated input features and input features from the 3D interpolation is visible in Fig. 5.

Table 7: The caption misses a description of the beta_x column. It has been added as follows:

"The precision-recall AUC (AUC-Pr.Re.) and the best average threshold value (beta_X) is shown."

A column with the corresponding beta_Ax values would be extremely relevant. Several studies analyzing the skill of hail algorithms use the Critical Success Index (CSI) and the HSS (Heidke Skill Score), which can be easily computed from the contingency table. It would be interesting to have those scores in Table 7 for the best beta_x and beta_Ax pair of each score (+ those of the ConvNet). beta_Ax, CSI and HSS columns were added to Table 7. As a

result, a description of CSI and HSS has been added in the description of scores in section 3.3.

L561: What do you mean by local here? The term "local" has been removed for clarity:

"The number of false alarms for the best ConvNet, i.e. the ConvNet with the highest AUC-ROC at a discrimination probability, [...]"

L565: Again, what is the meaning of local in this context? Same as the previous comment:

"The performance in terms of PSS for the [...]"

L565: Why is the beta_x value for MESH_95 = 33 mm in the text and 31 mm in Table 7? 31mm is the average among the 5 best variants of MESH_95. 33mm is the best variant.

L578: "varies significantly depending on the hail class." Why don't you show the results for each class instead of the average? It only adds two lines to the table and the reader will have the complete information. Done.

L606: It's not clear what accordingly refers to. You could write "...the threshold value is optimized". Modified to: "equivalent skill for severe hail detection on the test set if the threshold value is optimized"

L611: Do you have the predictions of MESH_95 in this case for comparison? As this is an important feature of the ConvNet, it guess that it should be similar. The following figures show the MESH_95 output for the event in Fig. 15 after different feature thresholds (beta_x): 0 mm, 20 mm, and 33mm (the best beta_x).

Maximum MESH_95 with a threshold beta_X = 0 mm

[Figure]

Maximum MESH_95 with a threshold beta_X = 20 mm

[Figure]

Maximum MESH_95 with a threshold beta_X = 33 mm

[Figure]

Figure 14: For the two POH and POSH, why don't you show the full range of thresholds up to 100%? This is interesting information. Change the y-axis ticks labels to a multiple of 10 for readability. Done, it now ranges from 0% to 100% in Fig. 14.

L638: "utilising the radar information of a unique timestep.": This should be mentioned in the method section (see my previous remark for L328). See comment L328.

Figure 15: On the bottom right there are two circles without grey lines. What are they? New cells that formed at 19:00 UTC, at the end of the time window.

L646: Did you look at other hail events in detail or only this one? What were the results? This is out of curiosity and doesn't need to be included in the paper. The event of the 11th of July is the event in the test dataset that had the most severe hail reported. It was the first to be looked at for that reason. No other events were analysed in such detail in the scope of this study, mainly because of time restrictions.

**Technical corrections**

L44: references should be sorted by year of publication. Done

L59: add "radar variables" after dual-polarization Done

L62: references should be sorted by year of publication. Done

L77: replace "traction" by attraction Done

L77: "In the work of Wang et al. (2018), they developed a CNN applied to…" → Wang et al. (2018) applied a CNN to… Done

L80: "In the work of Shi et al. (2020), they tracked…" → Shi et al. (2020) tracked… Done

L84: Ackermann et al. (2024) trained… Done

L97: "state-of-the-art"… hail detection methods?` Done

Table 1: I would write explicitly the 3 lowest angles on each row to avoid confusion (even if they are the same). Done, see Tab. 1.

Figure 1 caption: in the pdf, ESWD reports appear in grey-blue instead of grey, whereas reports from the Météo-France app appear in grey instead of black. It has been corrected accordingly in the title of Fig. 1:

"Hail reports between 2018 and August 2023 from the ESWD (grey-blue), the hailpad network of the ANELFA (orange) and the mobile application of Météo-France (small grey dots).

L104: "the three upper elevation angles". Done

L106: "corrected for attenuation". Done

L153: hailpad in one word. Done

L223: between Done

Figure 7: The label of the grey contour is not visible, consider using another color (e.g. red). Use the same scale for both colorbars for direct comparison. The max value for rain or small hail cases is higher than for severe hail. The grey contour was changed to red for visibility..

Table 5: True Negative Done

L457: decreasing instead of increasing? Indeed, it was corrected.

L458: Define ResNet18 on L384. See L384

L522: sensitive instead of sensible. Done

L531: remove rho_HV. Done

L548: State of the art is subjective. Replace with an objective word, e.g.: "Comparison of CNN with hail proxies/algorithms". Change to proxies.

L551: high instead of strong. Done

Figure 15: the green used severe hail reports appear dark and difficult to see. You could use the same green as in Fig.3 for the reports. A lighter green for the triangles has been used instead.